# The transcriptomic and epigenetic map of vascular quiescence in the continuous lung endothelium

Katharina Schlereth[1,2]*, Dieter Weichenhan[3], Tobias Bauer[4], Tina Heumann[1,2], Evangelia Giannakouri[1,2], Daniel Lipka[3], Samira Jaeger[5], Matthias Schlesner[4,6], Patrick Aloy[5,7], Roland Eils[4,8,9], Christoph Plass[3,10]†*, Hellmut G Augustin[1,2,10]†*

[1]European Center for Angioscience (ECAS), Medical Faculty Mannheim, Heidelberg University, Heidelberg, Germany; [2]Division of Vascular Oncology and Metastasis, German Cancer Research Center (DKFZ-ZMBH Alliance), Heidelberg, Germany; [3]Division of Epigenomics and Cancer Risk Factors, German Cancer Research Center, Heidelberg, Germany; [4]Division of Theoretical Bioinformatics, German Cancer Research Center, Heidelberg, Germany; [5]Joint IRB-BSC-CRG Program in Computational Biology, Institute for Research in Biomedicine (IRB Barcelona), The Barcelona Institute for Science and Technology, Barcelona, Spain; [6]Bioinformatics and Omics Data Analytics, German Cancer Research Center, Heidelberg, Germany; [7]Institució Catalana de Recerca i Estudis Avançats, Barcelona, Spain; [8]Institute of Pharmacy and Molecular Biotechnology, Heidelberg University, Heidelberg, Germany; [9]Bioquant Center, Heidelberg University, Heidelberg, Germany; [10]German Cancer Consortium, Heidelberg, Germany

*For correspondence:
schlereth@angiogenese.de (KS);
c.plass@dkfz.de (CP);
augustin@angiogenese.de (HGA)

†These authors contributed equally to this work

**Abstract** Maintenance of a quiescent and organotypically-differentiated layer of blood vessel-lining endothelial cells (EC) is vital for human health. Yet, the molecular mechanisms of vascular quiescence remain largely elusive. Here we identify the genome-wide transcriptomic program controlling the acquisition of quiescence by comparing lung EC of infant and adult mice, revealing a prominent regulation of TGFß family members. These transcriptomic changes are distinctly accompanied by epigenetic modifications, measured at single CpG resolution. Gain of DNA methylation affects developmental pathways, including NOTCH signaling. Conversely, loss of DNA methylation preferentially occurs in intragenic clusters affecting intronic enhancer regions of genes involved in TGFβ family signaling. Functional experiments prototypically validated the strongly epigenetically regulated inhibitors of TGFβ family signaling SMAD6 and SMAD7 as regulators of EC quiescence. These data establish the transcriptional and epigenetic landscape of vascular quiescence that will serve as a foundation for further mechanistic studies of vascular homeostasis and disease-associated activation.

DOI: https://doi.org/10.7554/eLife.34423.001

## Introduction

Endothelial cells (EC) are long-lived cells of the mesodermal lineage that line the inside of all blood vessels forming a single layer of organotypically-differentiated cells (*Augustin and Koh, 2017*; *Deanfield et al., 2007*; *Rafii et al., 2016*; *Regan and Aird, 2012*). Unlike unidirectional differentiation programs of short-lived epithelial cells, EC acquire a quiescent state during the transition to adulthood from which they can switch back to the activated phenotype resulting in disease-associated angiogenesis (*Hobson and Denekamp, 1984*; *Marcelo et al., 2013*). In fact, the maintenance

**eLife digest** The vascular system is made up of vessels including arteries, capillaries and veins that carry blood throughout the body. The inner surfaces of these blood vessels are lined with a thin layer of cells, called endothelial cells, which form a barrier and a communicating interface between the circulation and the surrounding tissue.

Early in an organism's life, when the vascular system is still growing, endothelial cells increase in number by dividing into more cells. In adulthood, as the vascular system reaches its full size, the endothelial cells maintain a stable number. As a result, an adult's vascular system has a resting layer of endothelial cells that does not divide. This is known as vascular quiescence, and scientists know little about how the body achieves and maintains it.

To unravel the mechanisms controlling vascular quiescence, Schlereth et al. studied endothelial cells taken from blood vessels in the lungs of newborn and adult mice. By comparing all the genes present at both developmental stages, the changes of gene activity in these cells could be measured. The results showed that the activity of genes strongly correlated with so called epigenetic changes in the genes involved in vascular quiescence. These are DNA modifications that can alter the function of a gene without affecting its underlying sequence.

Two genes in particular (Smad6 and Smad7) appeared to play an important role in vascular quiescence. Their corresponding proteins, SMAD6 and SMAD7, inhibit another group of proteins (TGFβ family) important for cell growth. The results showed that the endothelial cells in adult mice produced more SMAD6 and SMAD7 than in young mice. Therefore, endothelial cells of adult mice stop to increase in number and to migrate.

For the first time ever, Schlereth et al. have provided an extensive comparative analysis of gene activity and epigenetic changes to study vascular quiescence. The findings open a new chapter of vascular biology and will serve as a foundation for future research into the mechanisms of vascular quiescence. Problems in maintaining a resting layer of cells may lead to vascular dysfunction, which is associated with a wide range of diseases, such as stroke, heart disease and cancer making it a leading cause of death. In future, scientists may be able to develop new treatments that target specific molecules to help the body achieve a resting blood vessel system.

DOI: https://doi.org/10.7554/eLife.34423.002

of the quiescent vascular phenotype is of pivotal importance for human health. Loss of this EC phenotype is a hallmark of vascular dysfunction and a commonality of diverse life-threatening diseases ranging from sepsis, atherosclerosis to cancer (*Carmeliet and Jain, 2011*; *Gimbrone and García-Cardeña, 2016*). Likewise, major and socioeconomically important chronic diseases are characterized by a loss of the quiescent EC phenotype, like age-dependent macular degeneration and diabetic retinopathy. Vascular targeted therapies for such conditions are mostly aimed at interfering with EC activation programs (*Krzystolik et al., 2002*; *Tanabe et al., 2017*). Therefore, it is crucial to get a better understanding of the pathways involved in acquisition of vascular quiescence to improve vessel targeted therapies.

While the mechanisms of sprouting angiogenesis, network formation and remodeling have been molecularly unraveled in increasing detail, the molecular mechanisms of acquisition of EC quiescence are still poorly understood (*Adams and Alitalo, 2007*; *Herbert and Stainier, 2011*; *Jain, 2003*; *Korn and Augustin, 2015*). EC quiescence is an active process, as it not only involves the absence of activators but also microenvironmental factors as well as cell intrinsic mechanisms. Matricellular factors, contact with pericytes and hemodynamic forces act as microenvironmental determinants of vascular maturation (*Armulik et al., 2011*; *Baeyens et al., 2016*; *Davis et al., 2015*; *Stratman et al., 2017*; *Yousif et al., 2013*). To switch from an activated to a resting phenotype, numerous signaling pathways in EC including VEGF, NOTCH, FGF, TGFβ, angiopoietin and semaphorin signaling need to be properly regulated (*Marcelo et al., 2013*; *Patel-Hett and D'Amore, 2011*). To add another layer of complexity, there is an intricate crosstalk between these pathways and their integration is not completely understood (*Ehling et al., 2013*; *Holderfield and Hughes, 2008*; *Kim et al., 2011*). Furthermore, while some pathways are described to have clear pro- or anti-angiogenic effects, the impact of other signaling pathways is rather contextual or even still

controversially discussed. For instance, TGFß family signaling has been described to exert both, pro- and anti-angiogenic functions and to contextually cooperate with NOTCH signaling in the regulation of angiogenesis (*Cai et al., 2012*; *Dyer et al., 2014*; *Goumans et al., 2003*; *Goumans et al., 2002*; *Larrivée et al., 2012*; *Mallet et al., 2006*; *Mouillesseaux et al., 2016*; *Suzuki et al., 2008*). Thus, better molecular understanding of the way crucial EC-associated signaling pathways are regulated during the post-angiogenic acquisition of vessel quiescence will yield better mechanistic insights into vascular function during health and disease.

The mechanisms of EC activation during health and disease have been studied in substantial detail on the expression level of individual genes or pathways as well as genome-wide, for example genome-wide screens of regulators of angiogenesis or inflammation (*Cannon et al., 2013*; *del Toro et al., 2010*; *Massaro et al., 2015*; *Zhang et al., 2012*). Yet, no unbiased genome-wide systems biological effort from in vivo isolated EC has been made so far to identify the transcriptomic signatures associated with the acquisition and maintenance of EC quiescence. DNA methylation acts in concert with other epigenetic marks in defining chromatin states which create gene active and silenced compartments in the genome (*Jones, 2012*; *Soshnev et al., 2016*). Moreover, considering the increasingly recognized important roles of epigenetic alterations in the control of cellular differentiation processes (*Boland et al., 2014*; *Cabezas-Wallscheid et al., 2014*; *Lipka et al., 2014*), crucial molecular mediators of vessel maturation and EC quiescence may also be regulated by epigenetic mechanisms, which in EC have similarly not been studied in an unbiased genome-wide approach. The present study consequently aimed at obtaining and comparing transcriptomic and epigenomic signatures of EC from infant and young adult mice in order to identify most prominently regulated signaling pathways of vascular quiescence.

## Results

### Endothelial transcriptomic changes during acquisition of vascular quiescence

To identify factors and pathways regulated during acquisition of endothelial cell (EC) quiescence in an unbiased genome-wide manner, RNA-seq from freshly FACS-sorted lung EC isolated from infant (p8-p10) and adult mice (8–12 weeks) was performed (*Figure 1A* and *Figure 1—figure supplement 1A*). The purity of cell sorting was controlled by the validation of marker gene expression (*Figure 1—figure supplement 1B*). The phenotypic shift in marker expression (CD31/PECAM1 and CD34) between infant and young adult EC reflected the size of the corresponding EC (*Figure 1—figure supplement 1C*). EdU incorporation assays verified the almost complete arrest of cell proliferation in adult EC (*Figure 1B* and *Figure 1—figure supplement 1D*). Comparative transcriptome analysis identified 2,216 differentially expressed genes with pronounced regulation of genes mediating the induction of immune mechanisms, metabolism and phagosome pathways as well as the repression of genes controlling cell cycle, developmental angiogenesis, extracellular matrix interactions and cell contraction (Reactome-based analysis) (*Figure 1C*). To gain further insight into the mechanisms of EC quiescence, the protein interaction network (interactome) of the differentially regulated genes was generated (*Figure 1—figure supplement 2A*). The analysis of this interactome led to the identification of several pathways presumably relevant during the acquisition of EC quiescence (*Figure 1—source data 1*). Importantly, it resulted in further specification of immune system regulation, as JAK-STAT signaling was most significantly enriched (p=5.7×10⁻¹¹) in an EC quiescence network with most factors being induced, putatively mediating the repression of the pro-angiogenic growth factors *Fgf7* and *Fgf10* (*Figure 1—figure supplement 2B* and *Figure 1—source data 1*). Further network clustering identified densely connected protein communities within the interactome (*Figure 1—figure supplement 2C*). Functional annotation of the genes within each cluster revealed unique (SEMA, TGFB and ERBB signaling) enriched biological functions compared to Reactome-based analysis. Altogether, interactome analysis confirmed and extended the functional annotation of the genes differentially regulated during acquisition of EC quiescence.

In line with the EdU assays, prominent regulators of cell cycle arrest were validated to be downregulated during acquisition of EC quiescence in lung EC (*Figure 1D* and *Figure 1—source data 2*) as well as in brain and heart EC (*Figure 1—figure supplement 1E*). Altogether, these data suggest

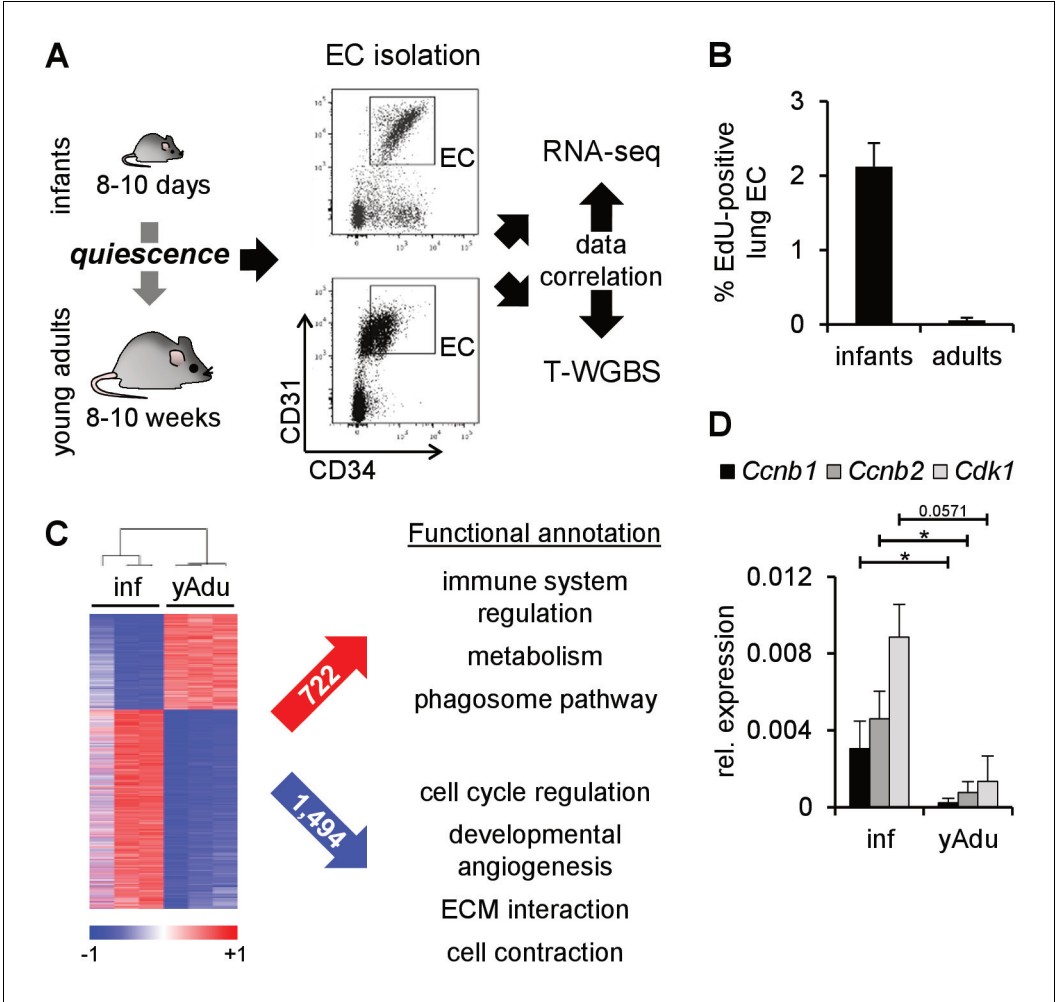

**Figure 1.** Transcriptomic changes of endothelial cells during acquisition of quiescence. (**A**) Scheme depicting the workflow of lung EC isolation by FACS from infant (inf) and young adult (yAdu) mice followed by transcriptome profiling by RNA-sequencing and DNA methylome analysis by tagmentation-based WGBS (T-WGBS). Shown are representative FACS profiles for the two different endothelial cell populations. (**B**) Analysis of lung EC proliferation *in vivo* by EdU assay. EdU was injected i.p. and lung EC were analyzed by FACS 17 hr later. n = 3; mean ± SD. (**C**) Hierarchical clustering of differentially expressed genes and functional annotation by Reactome overlap analysis of the 722 induced and 1,494 repressed genes regulated during the acquisition of EC quiescence. (**D**) Validation of the differential expression of cell cycle genes in infant and young adult lung EC using qPCR. n ≥ 3; mean ± SD; *p<0.05; Mann-Whitney Test.

DOI: https://doi.org/10.7554/eLife.34423.003

The following source data and figure supplements are available for figure 1:

**Source data 1.** Functionally enriched KEGG pathways Identified for (i) seeds in the EC maturation network, (ii) proteins in the EC maturation network as well as (iii) for all 3223 seeds.
DOI: https://doi.org/10.7554/eLife.34423.006

**Source data 2.** Cell cycle genes regulated in EC during the transition to a quiescent state.
DOI: https://doi.org/10.7554/eLife.34423.007

**Figure supplement 1.** Characterization of endothelial cell populations.
DOI: https://doi.org/10.7554/eLife.34423.004

**Figure supplement 2.** The quiescence-dependent interactome of lung EC.
DOI: https://doi.org/10.7554/eLife.34423.005

that the comparison of the transcriptomic profile of infant and adult EC displayed valid quiescence-dependent changes.

## TGFß family signaling is inhibited in quiescent EC

Corresponding to the arrest of EC proliferation, the transcriptomic screen identified distinct changes of angiogenic molecule expression. Sixteen pro-angiogenic molecules were down-regulated and eight anti-angiogenic molecules were up-regulated during acquisition of vascular quiescence (*Figure 2A*). This involved the downregulation of angiogenic signal transducing cell surface receptors (*Tgfbr2, Bmpr1, Notch3* and *Fgfr1*) as well as the upregulation of quiescence signal transducing receptors, most notably *Cd36* and the angiopoietin receptor *Tek* (encoding for Tie2 protein), probably the functionally best characterized maturation receptor of EC (*Augustin et al., 2009*). Several autocrine and paracrine acting secreted angiogenic molecules were identified as differentially

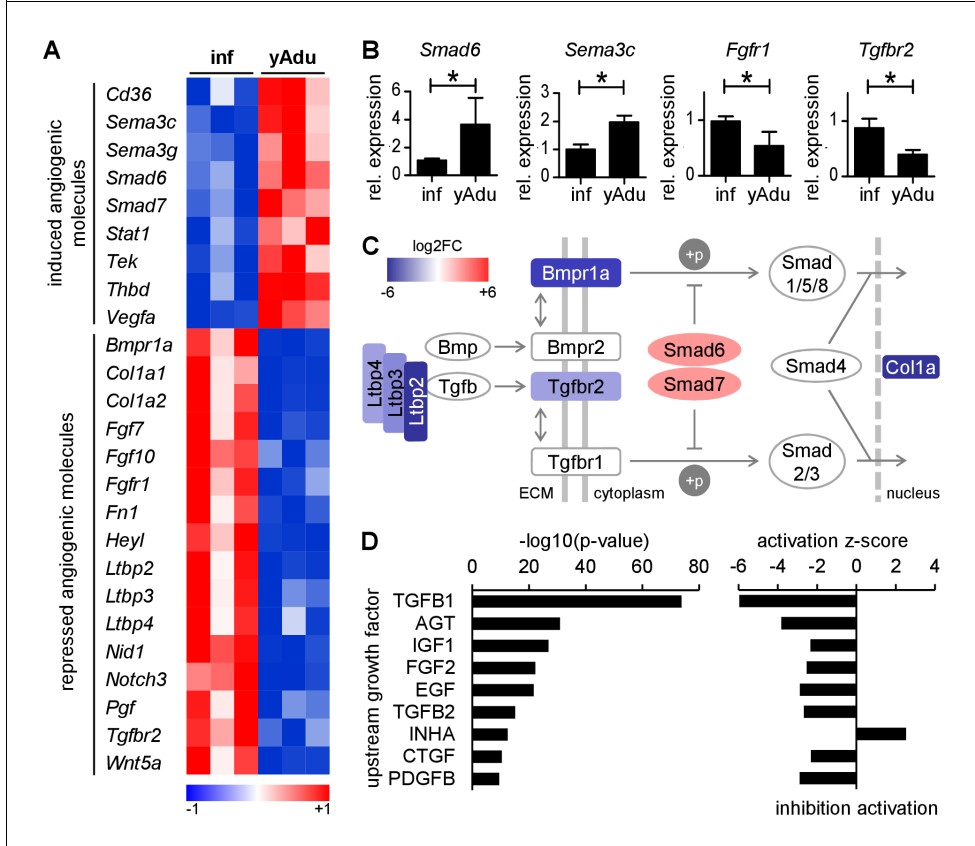

**Figure 2.** TGFß family signaling is inhibited in quiescent EC. (**A**) Heatmap depicting the expression of anti- and pro-angiogenic molecules during the acquisition of vascular quiescence. Depicted are significantly regulated genes (p<0.05) that are differentially expressed by at least 1.75 fold. (**B**) Validation of the expression pattern of anti- and pro-angiogenic molecules in lung EC by qPCR in independent biological samples. n ≥ 3; mean ±SD; *p<0.05; Mann-Whitney Test. (**C**) Schematic representation of the regulation of TGFβ family signaling molecules during acquisition of vascular quiescence emphasizing the overall inhibition of this pathway. Shown by color-code is the gene regulation of individual TGFβ family pathway members (red, upregulated; blue, downregulated; log2FC: log2 fold change). Ltbp: latent TGF-beta binding protein; Bmpr1a: Bone Morphogenetic Protein Receptor Type 1A/Alk3; Bmpr2: Bone Morphogenetic Protein Receptor Type 2; Tgfbr2: Transforming Growth Factor Beta Receptor 2; Tgfbr1: Transforming Growth Factor Beta Receptor 1/Alk5; Smad6 and Smad7: inhibitory SMADs; Smad1/5/8 and Smad2/3: receptor-regulated SMADs; Smad4: Co-SMAD; Col1a: TGFß target gene Collagen type I alpha. (**D**) Upstream growth factor analysis with activation state prediction of all differentially expressed genes during transition to EC quiescence. Depicted are the transformed (-log10) p-value and the predicted activation state (z-score) of the upstream growth factor according to Ingenuity Pathway Analysis. See methods section for further details. The following figure supplements are available for *Figure 2*.

DOI: https://doi.org/10.7554/eLife.34423.008

The following figure supplement is available for figure 2:

**Figure supplement 1.** Analysis of vascular quiescence-dependent gene expression.
DOI: https://doi.org/10.7554/eLife.34423.009

expressed: up-regulation of the semaphorins *Sema3C* and *Sema3G* as well as thrombomodulin (*Thbd*) and downregulation of *Fgf7*, *Fgf10*, placental growth factor (*Pgf*) and *Wnt5a*. Comparative quantitative PCR (qPCR) of selected candidate molecules validated gene expression changes identified by RNA-seq confirming the high robustness of the sequencing data (*Figure 2B* and *Figure 2— figure supplement 1A*).

Among these angiogenesis-regulating molecules, the TGFβ pathway was identified as the most concordantly downregulated signaling pathway with repression of matrix-associated latent TGFβ-binding proteins (*Ltbp2*, *Ltbp3*, *Ltbp4*) and receptors (*Tgfbr2*, *Bmpr1*) as well as upregulation of the inhibitory Smads, *Smad6* and *Smad7* (*Figure 2C*). Correspondingly, upstream analysis of the complete set of differentially expressed genes including activation state prediction of the corresponding upstream regulator revealed TGFß1 as the most significantly inhibited growth factor (*Figure 2D*). Collectively, the data uncovered complex transcriptional changes of angiogenic molecules governing EC quiescence. They highlight a series of pathways that have all been associated with EC activation, most notably the TGFβ pathway.

## Vascular quiescence is accompanied by a prominent loss of DNA methylation at intronic enhancer regions

To assess the impact of epigenetic remodeling in the vascular system during acquisition of vascular quiescence, we analyzed the expression of chromatin modifying enzymes (CME) (*Plass et al., 2013*). About 10% of CMEs were differentially expressed with the majority being repressed in adult EC (*Figure 2—figure supplement 1B and C*), including the *de novo* DNA methyltransferase *Dnmt3a* and four additional DNA methylation modifiers (*Gadd45a*, *Tet1*, *Fos*, *Uhrf1*) (*Figure 2—figure supplement 1D*). This suggested that the patterns of DNA methylation were more dynamic in infant compared to adult EC. It is now accepted that the DNA methylome gives a precise picture of the epigenetic state, defining active and inactive regions within genomes (*Cabezas-Wallscheid et al., 2014*; *Jones, 2012*; *Lipka et al., 2014*; *Soshnev et al., 2016*). Consequently, tagmentation-based whole genome bisulfite sequencing (T-WGBS) of biological triplicates of infant and young adult EC samples enabled the identification of quiescence-dependent DNA methylation changes at single CpG nucleotide resolution (*Figure 1A* and *Figure 3—figure supplement 1A*). T-WGBS data were highly reproducible in MassARRAY using independent biological replicates (*Figure 3—figure supplement 1B and C*). Bioinformatic analysis revealed a significant increase of CpGs with >80% methylation level in adult EC compared to infant EC (p=$6.87 \times 10^{-6}$) demonstrating a global increase in CpG methylation during the transition to vascular quiescence (*Figure 3A*). In total, DNA methylation changes resulted in more than 1.4M differentially methylated CpGs (DMCs). Retaining only the consecutive DMCs (68,184) with a 10% methylation difference showed a significant intragenic enrichment (*Figure 3—figure supplement 1D*). These DMCs gave rise to 18,333 differentially methylated regions (DMRs) (*Figure 3B*). The distribution of methylation differences among the DMRs demonstrated a more prominent loss (20–60% methylation difference between infants and young adult) than gain of methylation (10–40% methylation difference) (*Figure 3C*). Applying a threshold of 10% methylation difference, young adults acquired a gain of methylation in 54% of DMRs and a loss of methylation in 46% of DMRs during acquisition of quiescence (20% gain and 80% loss, respectively, when applying 30% as threshold) (*Figure 3B*). Both, loss and gain of methylation DMRs were depleted on transcription start sites (TSS), but enriched up- and downstream (10–100 kb distant) from the TSS (*Figure 3D*). There was an overall significant enrichment of DMRs in introns (OR 2.65, p<0.0001), most prominent in intron 1–3 of a given gene (*Figure 3E* and *Figure 3—figure supplement 1E*). Notably, hypomethylated DMRs were overrepresented in introns and underrepresented in intergenic regions (*Figure 3—figure supplement 1F*). Overlap of the DMRs with lung-identified genomic regulatory features (*Shen et al., 2012*; *Yue et al., 2014*) further specified the DMRs to coincide with putative enhancer regions (H3K4me1 positive regions, enhancer-promoter units [EPU], DNase hypersensitive regions) but not with transcription start sites (CpG islands, TSS defined by FANTOM consortium, H3K4me3 positive regions) (*Figure 3F*). This could be visualized in an exemplary way in genome browser views depicting CpG islands and putative enhancer regions together with the identified DMRs (*Figure 3G* and *Figure 4E*). Essentially, DNA methylation profiling suggested a prominent loss of methylation at intronic enhancers during acquisition of quiescence in lung EC.

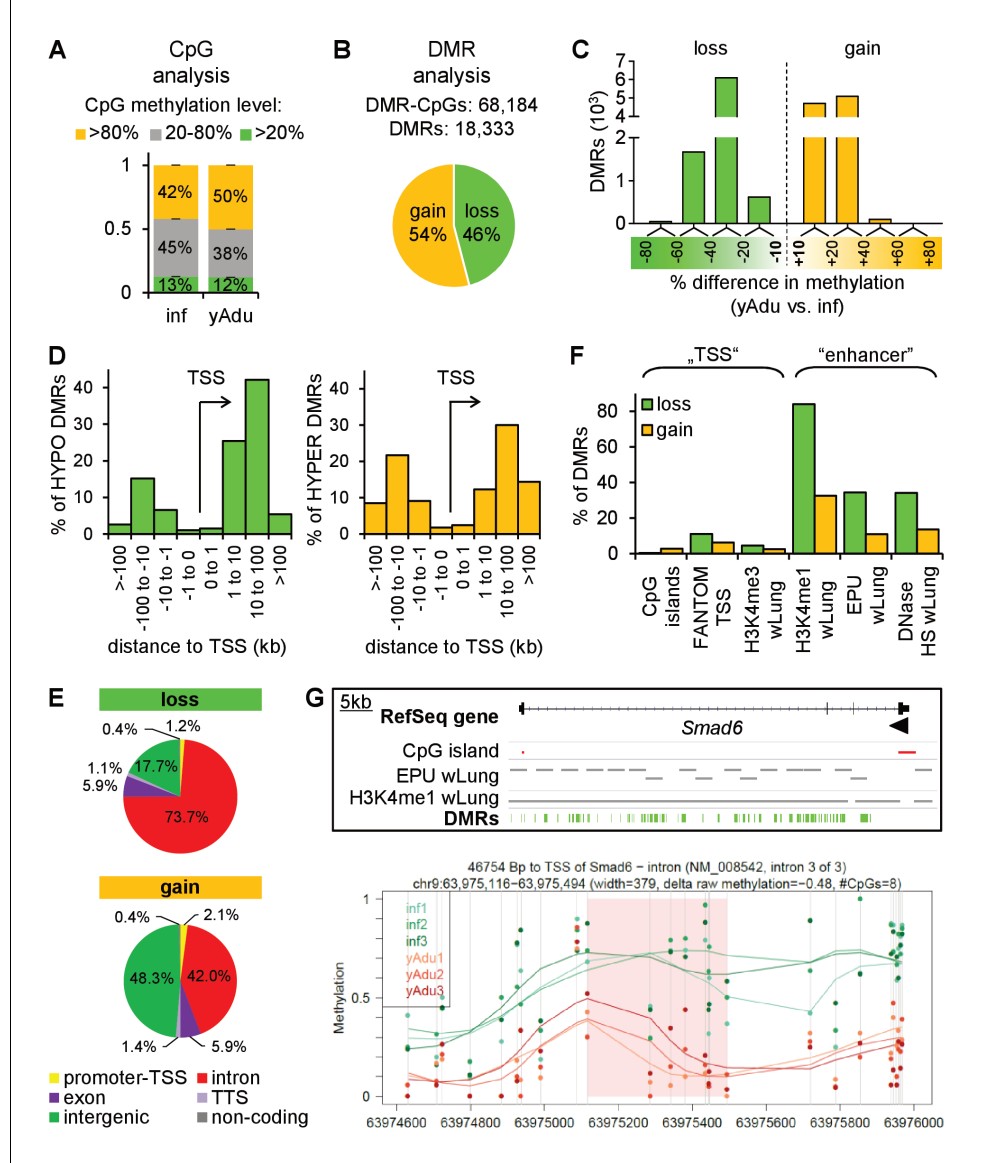

**Figure 3.** Vascular quiescence is accompanied by a prominent loss of DNA methylation at intronic enhancer regions. (**A, B**) Data of tagmentation-based WGBS of infant and young adult lung EC were analyzed on the level of single CpGs (**A**) and differentially methylated regions (DMRs) (**B**). (**C**) Distribution of the methylation difference (in %) of DMRs during the acquisition of EC quiescence. Depicted are all changes ≥ 10% between young adult and infant EC. (**D**) Localization of hypo- (left) and hyper- (right) methylated DMRs relative to the transcription start site (TSS) in kb. (**E**) Genomic location of the hypo- (top) and hyper- (bottom) methylated DMRs. TTS = transcription termination site. (**F**) Overlap analysis of DMRs with known regulatory DNA elements of the lung. wLung = whole lung tissue, EPU = enhancer promoter units, HS = hypersensitivity. (**G**) Top: Genome browser view of the *Smad6* gene locus. Green bars represent the DMRs showing significant loss of methylation during the acquisition of vascular quiescence. Arrowhead represents the transcription start site. Grey bars represent putative enhancer regions previously defined. Bottom: Detailed view of one representative DMR (transparent red) showing the raw (dots) and the smoothed (lines) methylation data of EC from three infant (green) and three young adult (red) mice. CpGs are represented by vertical lines. The following figure supplements are available for *Figure 3*.

DOI: https://doi.org/10.7554/eLife.34423.010

The following figure supplement is available for figure 3:

**Figure supplement 1.** DNA methylation profiling of lung EC during transition to vascular quiescence.
DOI: https://doi.org/10.7554/eLife.34423.011

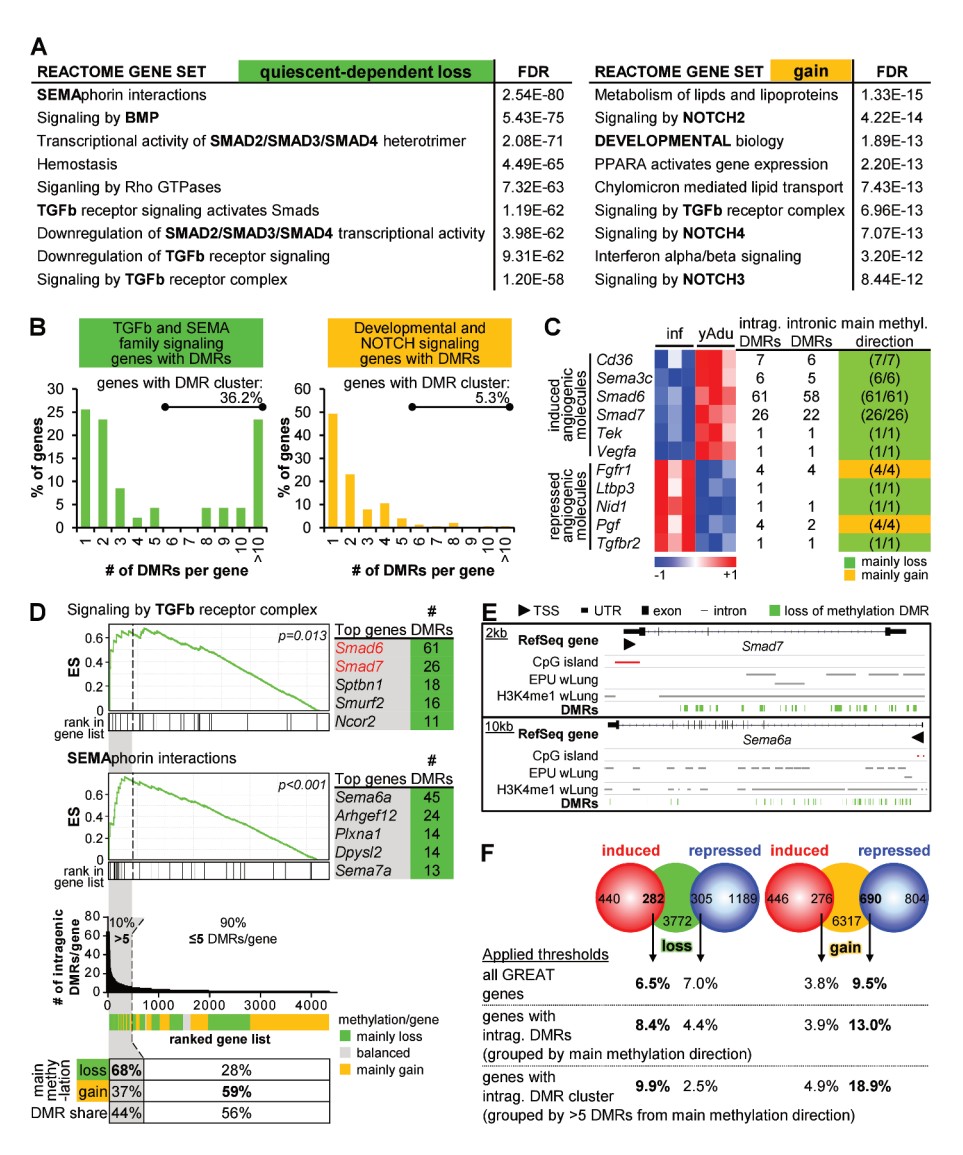

**Figure 4.** Clustering of loss of methylation DMRs in genes of TGFβ and semaphorin signaling. (**A**) Functional annotation of genes next to loss of methylation DMRs by the GREAT tool. Listed are the top Reactome gene sets according to the FDR value (sorted in ascending order). (**B**) Distribution of DMR number/gene (with percentage of genes containing DMR clusters). (**C**) Characteristics of differentially expressed angiogenic molecules during vascular quiescence that were also differentially methylated (see **Figure 2A**). (**D**) Ranking of genes according to the number of intragenic DMRs. The subset of genes containing >5 DMRs (=DMR cluster) is highlighted in grey. Top: Enrichment analysis of the ranked list of genes demonstrated an overrepresentation of semaphorin and TGFβ signaling genes among the genes containing DMR cluster. Red genes: expression significantly induced >1.75 fold, p<0.05. ES = enrichment score. Bottom: Methylation direction and DMR distribution among genes with >5 or≤5 DMRs/gene, respectively. (**E**) Genome browser views of gene loci with large DMR clusters (*Smad7, Sema6a*). Putative enhancer regions are shown as grey bars (EPU = enhancer promoter units; wLung = whole lung tissue). (**F**) Direct correlation of significantly differentially expressed (red and blue circles) and methylated (green and orange circles) genes applying different thresholds to the number of DMRs per gene: no limitations, only genes with intragenic DMRs and only genes with DMR cluster. Numbers in circles represent the overlap without thresholding the number of DMRs. Percentages refer to the fraction of differentially methylated genes that overlap with differentially expressed genes. The following figure supplements are available for **Figure 4**.

DOI: https://doi.org/10.7554/eLife.34423.012

The following figure supplement is available for figure 4:

*Figure 4 continued*

**Figure supplement 1.** Correlation of gene expression and DNA methylation profiles of endothelial cell during the acquisition of quiescence.

DOI: https://doi.org/10.7554/eLife.34423.013

## Clustered intragenic loss of methylation regulates TGFβ family and semaphorin signaling in quiescent endothelial cells

Methylome analysis identified strong DNA demethylation of putative intronic enhancer regions suggesting that DMRs may have regulatory potential on gene expression. As enhancer regions could influence the expression of distant genes, up to three nearby genes were assigned to the quiescence-dependent DMRs by applying the GREAT tool to identify genes likely affected by differential methylation (*McLean et al., 2010*). The same tool was also used to identify significantly enriched biological functions among the genes next to quiescence-dependent regulated DMRs. The functional annotation of genes next to hypomethylated DMRs revealed signaling pathways controlling vascular patterning (semaphorin interactions) and development/homeostasis (TGFβ family signaling) (*Figure 4A*). Notably, 94% of the hypomethylated DMRs affecting genes of these pathways were located in intragenic regions with eight DMRs per gene on average (*Figure 4—figure supplement 1A*, left) resulting in DMR clustering (more than five DMRs per gene locus) in more than 36% of the affected genes (*Figure 4B*). Contrary, 56% of the hypermethylated DMRs were located in intragenic regions with on average two DMRs per gene (only 5.3% of genes harbored five or more hypermethylated DMRs). Furthermore, genes in the vicinity of gain of methylation DMRs were annotated as endothelial fate control genes (NOTCH signaling) (*Figure 4A and B*, *Figure 4—figure supplement 1A* right). In summary, in line with transcriptome analysis, genes of TGFß family and semaphorin signaling were similarly affected by differential DNA methylation. Apart from the functional annotation of genes located in the vicinity of DMRs, these analyses revealed an overrepresentation of DMRs within certain gene loci. Notably, this clustering of DMRs was mostly restricted to loss of methylation DMRs. Correspondingly, DMR clustering was most obvious in induced angiogenic molecules (*Figure 4C*).

The ranking of all genes according to the number of intragenic DMRs further demonstrated that (i) 10% (446) of the affected genes contained more than five intragenic DMRs corresponding to 44% (5,180) of all intragenic DMRs and (ii) 68% of the clustered DMRs showed loss of methylation (*Figure 4D*, bottom). This analysis highlighted the existence of large intragenic DMR clusters that undergo concerted differential DNA methylation in about 450 genes during transition to endothelial cell quiescence. Furthermore, enrichment analysis among this ranked list of genes confirmed that DMR clusters were overrepresented in genes of TGFβ and semaphorin signaling, particularly in *Smad6*, *Smad7* and *Sema6a* with up to 61 DMRs in one gene (*Figures 3G 4D*, top, and *4E*).

The correlation of quiescence-related expression and methylation profiles by gene set enrichment analysis (GSEA) revealed an enrichment of genes with hypomethylated DMRs (young adult vs. infant EC) among the genes induced in adult EC independent of the genomic loci of one or several DMRs. Conversely, genes with hypermethylated DMRs were overrepresented among the genes upregulated in infant EC (*Figure 4—figure supplement 1C*). In line with this, there was an unexpected high overlap of differentially expressed genes and genes affected by DNA methylation changes. Notably, we identified an increasing overlap of transcriptome and epigenome, when scoring genes on the basis of intragenic DMRs and even more so when scoring based on clustered DMRs (*Figure 4F*). This correlation implies that genes with intragenic clusters of loss of methylation DMRs are more likely to be induced during acquisition of vascular quiescence than repressed, while those genes with clustered gain of methylation DMRs are more likely to be repressed.

Together, these data suggest that vascular quiescence is accompanied by a prominent loss of methylation in intragenic DMR clusters preferentially affecting and regulating the expression of genes involved in the regulation of TGFβ family, semaphorin and NOTCH signaling in this process.

## Epigenetically-regulated SMAD6 and SMAD7 control vascular quiescence

Both, the transcriptome and the methylome analysis had highlighted TGFβ family members as relevant molecules during vascular quiescence. Among these, *Smad6* and *Smad7* contained the largest

intragenic cluster of hypomethylated DMRs and showed significant induction during transition to quiescence in lung EC (*Figure 2B*, *Figure 2—figure supplement 1A* and *Figure 4B*). We therefore focused further functional experiments prototypically on these molecules. SMAD6 and SMAD7 inhibit TGFβ family signaling by different mechanisms (*ten Dijke and Arthur, 2007*). Corresponding to the observed lung endothelial upregulation of *Smad6* and *Smad7* during acquisition of quiescence, we detected a strong decrease in R-SMAD (receptor-regulated SMADs, SMAD2/3 and SMAD1/5/8) phosphorylation in adult lung tissue confirming an inhibition of TGFβ family signaling in adult mice (*Figure 5A*). The upregulation of *Smad6* and *Smad7* during acquisition of vascular quiescence was not restricted to the lung, but also detected in EC isolated from adult mouse brain and heart (*Figure 5B* and *Figure 5—figure supplement 1A*). Correspondingly, R-SMAD phosphorylation was decreased in these organs suggesting that TGFβ signaling may vessel bed independently act as global regulator of vascular quiescence (*Figure 5—figure supplement 1B and C*). As reduced signaling could also be mediated by decreased ligand levels, BMPs and TGFB1 were determined in different organs (*Figure 5—figure supplement 1D*). TGFβ family ligands were consistently higher in adult organs indicating that regulation of vascular quiescence is achieved through cell intrinsic blocking of the downstream signaling machinery, that is by downregulation of receptors and upregulation of inhibitory SMADs.

To further examine if the expression of SMAD6 and/or SMAD7 meditates resistance of EC towards ligand-dependent receptor activation, HUVEC were lentivirally transduced and analyzed in vitro. Forced expression of SMAD6 or SMAD7 in cultured EC, but not gene silencing, led to the reduction of SMAD2/3 and SMAD1/5/8 phosphorylation upon ligand stimulation (*Figure 5C and D* and *Figure 5—figure supplement 2A*). Cells overexpressing both, SMAD6 and SMAD7 showed reduced viability upon prolonged cell culture, which was confirmed by MTT assay (*Figure 5E* and *Figure 5—figure supplement 2B*). Proliferation and migration of EC overexpressing SMAD6 and SMAD7 was reduced confirming the limited angiogenic potential of SMAD6 and SMAD7 expressing cells (*Figure 5F and G* and *Figure 5—figure supplement 2C and D*). Together, these data validate based on expression in vivo and functional experiments in vitro the negative regulation of the TGFβ family signaling as critical for the acquisition of endothelial quiescence during development.

## Discussion

Loss of the quiescent EC phenotype is at the heart of vascular dysfunction that is associated with some of the most common and most devastating diseases in humans. Vascular quiescence and the switch from quiescent to activated EC therefore need to be tightly regulated. How the resting phenotype is achieved on the genome-wide epigenetic and transcriptomic level has not been studied so far. This study presents the first unbiased systems biology based profiling of primary EC isolated from infant and adult mice, revealing surprising transcriptomic and epigenetic signatures. It thereby establishes a roadmap for further molecular dissection of mechanisms of vascular quiescence. Specifically, extensive bioinformatic data analysis and functional validation demonstrated that the acquisition of EC quiescence (i) is associated with distinct transcriptomic changes affecting pro- and anti-angiogenic molecules with prominent down-regulation of TGFß family signaling molecules and (ii) is accompanied by gain of DNA methylation and more prominent loss of DNA methylation at intragenic enhancer regions preferentially affecting genes of TGFß family and semaphorin signaling. In general, the identified gene expression changes (iii) correlate with changes in DNA methylation in corresponding loci. Notably, (iv) most hypomethylated DMRs were concentrated in large intragenic clusters, with the gene loci of the inhibitory SMADs (*Smad6* and *Smad7*) among the genes containing the largest clusters. Cellular experiments (v) confirmed SMAD6 and SMAD7 as regulators of endothelial quiescence. Together, this study highlights the central role of suppression of TGFβ family signaling during acquisition of vascular quiescence and establishes a database enabling the datamining of transcriptomic and epigenetic changes associated with the acquisition of the quiescent EC phenotype.

Although the importance of epigenetic mechanisms regulating vascular function has long been recognized (*Dunn et al., 2015*; *Matouk and Marsden, 2008*; *Yan and Marsden, 2015*), studies of DNA methylation in EC are still in their infancy. Only few genome wide systems biology studies have been performed to dissect epigenetic changes in the vasculature and even less studies have systematically correlated epigenetic changes to transcriptomic signatures. It has been described that

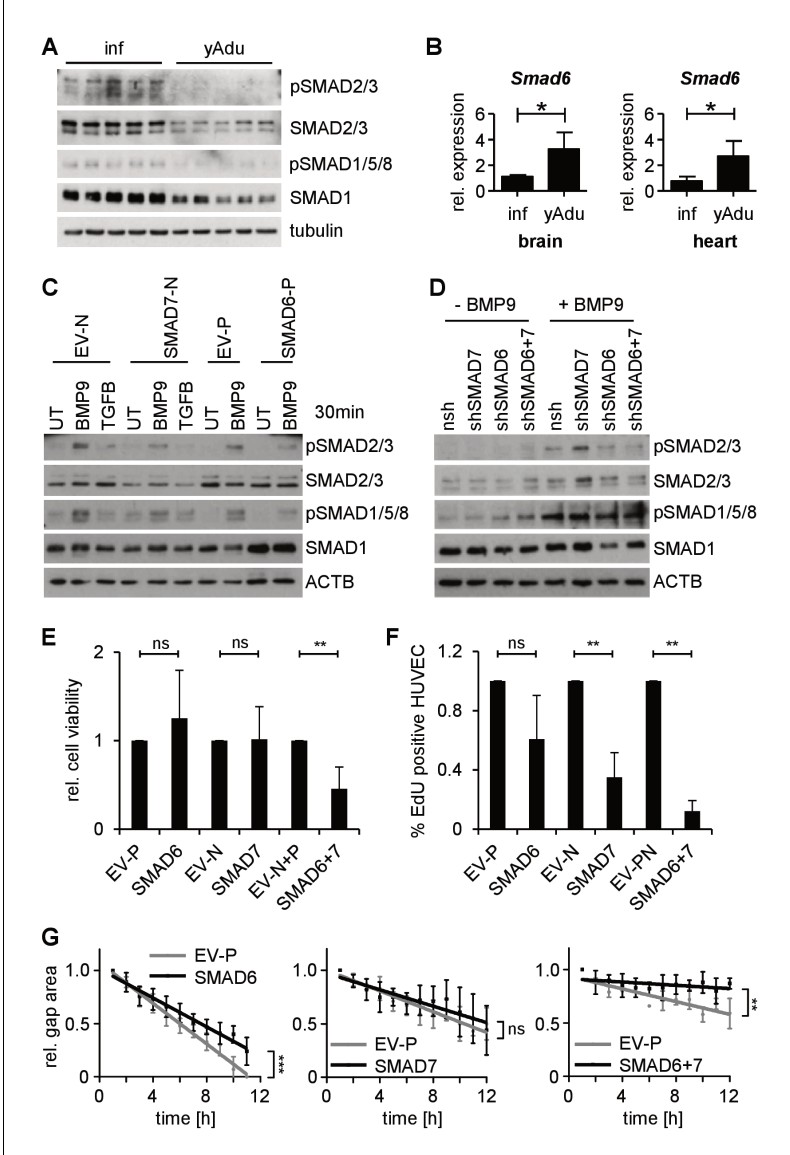

**Figure 5.** SMAD6 and SMAD7 control endothelial quiescence. (**A**) Immunoblot showing the analysis of TGFβ family signaling in whole lung lysates of infant (n = 5) and young adult (n = 5) mice. (**B**) Validation of *Smad6* induction during maturation in the brain and the heart by qPCR. n = 4; mean ±SD; *p<0.05; Mann-Whitney Test. (**C,D**) Immunoblots showing the analysis of TGFβ family signaling in HUVEC stably expressing (**C**) or with silenced (**D**) SMAD6 or SMAD7. UT = untreated, nsh = non silencing shRNA. (**E**) Analysis of cell viability by MTT assay. ns = non significant; **p<0.01; unpaired t-test. (**F**) Analysis of HUVEC proliferation by EdU incorporation assay (4 hr) followed by FACS analysis. n ≥ 3; ns = non significant; **p<0.01; unpaired t-test. (**G**) Analysis of HUVEC migration by lateral scratch wound assay. The gap was measured every hour. EV = empty vector, N/p=neomycin/ puromycin resistance. ns = non significant; **p<0.01; ***p<0.001; Linear Regression Analysis. The following figure supplements are available for *Figure 5*.

DOI: https://doi.org/10.7554/eLife.34423.014

The following figure supplements are available for figure 5:

**Figure supplement 1.** TGFß family signaling in different organs of infant and young adult mice.

DOI: https://doi.org/10.7554/eLife.34423.015

**Figure supplement 2.** Characterization of the role of SMAD6 and SMAD7 in endothelial cells.

DOI: https://doi.org/10.7554/eLife.34423.016

proximal promoter DNA methylation regulates EC gene expression and thereby controls EC fate (*Shirodkar et al., 2013*). Similarly, flow-dependent DNA methylation changes in vessel wall cells have been shown to be enriched in GC-rich, 5' untranslated, and exonic regions (*Davies et al., 2014*; *Jiang et al., 2015a*; *Jiang et al., 2015b*). These previous studies in EC have determined an enrichment of promoter-associated changes in DNA methylation that coincide with gene regulation. Here, we generated genome-wide high resolution DNA methylation data of primary infant and young adult EC which enabled the identification of an unusual abundance of intragenic (mainly intronic) demethylation events with numerous genes harboring large intronic DMR clusters. Although little is known about the impact of intragenic DNA methylation changes on gene expression, some studies suggest a role in alternative promoter usage or splicing (*Kulis et al., 2013*; *Maunakea et al., 2013*; *Maunakea et al., 2010*; *Shukla et al., 2011*). Conversely, the quiescent-dependent DMRs identified in the present study overlapped with putative intragenic enhancer features. Moreover, the genome-wide association with gene expression data revealed a good correlation of intragenic hypo-methylation and gene induction or hypermethylation and gene repression, respectively. The presence of clustered DMRs was even more likely associated with differential expression changes of the affected gene, suggesting that we have identified intronic regions with high regulatory potential during the acquisition of vascular quiescence. Notably, while we identified a more prominent loss than gain of methylation during acquisition of quiescence, atherosclerosis progression in human appears to be associated with a distinct gain of methylation signature (*Zaina et al., 2014*). Altogether, these data highlight the important role of epigenetic modifications in the regulation of EC functions and have led to the identification of distinct genomic regions with crucial regulatory potential that will serve as a foundation for further functional studies.

Among the transcriptomically and epigenetically regulated genes were molecules of pathways with well-established EC-regulatory function including NOTCH, semaphorin, FGF, WNT and TIE signaling (*Marcelo et al., 2013*; *Patel-Hett and DAmoreD'Amore, 2011*). Yet, both unbiased genome-wide approaches of this study, RNA-seq and T-WGBS, converged on a crucial role of TGFß family signaling in the regulation of vascular quiescence. This finding is not novel per se, but TGFß family signaling has been described to result in both, pro- and anti-angiogenic functions in a very context dependent manner (*Cai et al., 2012*; *Dyer et al., 2014*; *Goumans et al., 2003*; *Goumans et al., 2002*; *Larrivée et al., 2012*; *Mallet et al., 2006*; *Suzuki et al., 2008*). Here, we show in the physiologically most relevant vascular quiescence process (i.e. the comparison of infant and young adult EC) that many important signaling molecules of the TGFß family pathway are differentially regulated leading to reduced signal transduction via R-SMAD phosphorylation in adult tissue despite high ligand levels. As a decisive point, the inhibitory SMADs, *Smad6* and *Smad7*, were epigenetically inhibited by large intronic clusters of methylated DNA during early development and induced in adult EC. Cell culture based analysis confirmed that the expression of SMAD6 and/or SMAD7 in EC mediates resistance to ligand treatment thereby blocking TGFß family signaling. Interestingly, SMAD6 and SMAD7 have originally been identified as vascular SMADs (*Topper et al., 1997*). Their induction upon laminar shear stress highlights the importance of blood flow in maintaining the quiescent vascular phenotype. Although ubiquitously expressed, they exert particularly prominent vascular function. Smad6 has recently been described to act anti-angiogenic in sprouting assays (*Mouillesseaux et al., 2016*) and Smad7 has been shown to inhibit peritoneal angiogenesis (*Peng et al., 2013*). Correspondingly, genetic deletion of *Smad6* or *Smad7* leads to major cardiovascular defects with partial prenatal mortality (*Smad6*) (*Galvin et al., 2000*) or to massive growth retardation with reduced viability (*Smad7*) (*Tojo et al., 2012*). In conclusion, the present study underlines the central role of TGFß family pathways during acquisition of the resting EC phenotype including prominent regulation of inhibitory SMADs on the epigenetic level.

Remarkably, molecules of the VEGF pathway, the master regulator of angiogenesis induction, were both, transcriptionally and epigenetically not identified as being significantly differentially regulated during the transition to EC quiescence. Surprisingly, the ligand *Vegfa* was significantly higher expressed in adult EC compared to infant EC. High expression of VEGFA has also been described in the adult brain (*Licht and Keshet, 2013*). Autocrine VEGF has been proposed to be essential for EC survival thereby fulfilling homeostatic functions. It appears to serve as a rheostat of paracrine acting VEGF to possible even exert negative regulatory functions. In fact, analyses of genetically engineered mice specifically lacking EC-produced VEGF indicate that paracrine VEGF does not compensate for autocrine VEGF (*Lee et al., 2007*). Conversely, expression of the VEGF decoy receptor *Flt1/*

*Vegfr1* (*Boucher et al., 2017*; *Fong et al., 1995*; *Kappas et al., 2008*; *Kearney et al., 2002*) was in the present study significantly increased (but below threshold of 1.75x) during acquisition of quiescence. Since the expression of the primary VEGF signal transducing receptor *Kdr/Vegfr2* was neither in RNA-seq nor in qPCR validation data significantly changed, this results in an increased VEGFR1/VEGFR2 ratio in adults. Altogether, these data imply that VEGF signaling is not dominantly regulated during the transition to vascular quiescence but instead undergoes specific and situation-specific dynamic regulation.

In conclusion, the present study has for the first time established the transcriptomic and epigenetic landscape of vascular quiescence in primary lung EC that might also apply to EC of other tissues. These findings reveal that vascular quiescence is comprehensively regulated on both transcriptional and DNA methylation level converging on the prominent inhibition of TGFß family signaling in adult EC. Moreover, the acquisition of the resting EC phenotype is accompanied by a dynamic balance of activating and inhibiting pathways. These findings might in the future potentially pave the ways towards fundamentally novel strategies to therapeutically interfere with unwanted EC activation during disease. The study thereby lays a foundation for mechanistic and functional studies of individual genes and complex gene signatures involved in the transition from the activated to the quiescent phenotype and vice versa.

## Materials and methods

### Mice

C57BL/6N female mice were obtained from Taconic (Taconic Biosciences, Europe). After arrival, mice were kept in 'Individually Ventilated Cages' (IVC) to maintain the health status as declared by Taconic as 'Murine Pathogen Free'. Lungs were processed for EC isolation 1–3 days after arrival. Animals had free access to food and water and were kept in a 12 hr light-dark cycle. All mice were handled according to the guidance of the Institute and as approved by the German Cancer research center (No. DKFZ305). No statistical methods were used to predetermine sample size. The experiments were not randomized and the investigators were not blinded to allocation during experiments and outcome assessment.

### Mouse endothelial cell isolation by FACS

Endothelial cell (EC) isolation was performed as described previously (*Korn et al., 2014*). In brief, mice were sacrificed and lung, heart and brain of infant or young adult mice were surgically removed and cut into small pieces. Tissue pieces were digested in Dulbecco´s Modified Eagle´s medium (DMEM, ThermoFisher Scientific, Germany) containing 1.25 mM $CaCl_2$, 200 U/ml Collagenase I (Sigma, Munich, Germany) and 10 µg/ml DNaseI (Roche, Germany) at 37°C for 45 min. Single-cell suspensions were prepared by passing the digestion mix through 18G and 19G cannula syringes and filtering through a 100 µm cell strainer. To remove the myelin from the brain cell suspension 22% BSA was added (1:1 v/v), and after centrifugation the upper myelin layer was discarded. Cells were stained for the negative markers PDPN, LYVE1, PTPRC and LY76 for 30 min at 4°C in PBS/5% fetal calf serum (FCS). Cells stained with negative markers were depleted by incubation with Dynabeads (ThermoFisher Scientific, Germany) for 30 min at 4°C on the rotator followed by magnetic removal. The remaining cells were positively stained with antibodies against CD31 and CD34 in PBS/5% FCS for 30 min at 4°C. Dead cells were excluded by phosphatidylinositol (PI) staining (1:2000). $PTPRC^-LYVE1^-LY76^-PDPN^-PI^-CD31^+CD34^+$ cells were sorted with a BD FACSAria ll (BD Biosciences, Heidelberg, Germany)

The following primary antibodies were used: rat anti-CD31 (551262, BD Biosciences, diluted 1:100), rat anti-CD34 (48–0341, eBioscience, diluted 1:50), rat anti-LY76 (561032, BD Biosciences, diluted 1:200), rat anti-LYVE1 (53–0443, eBioscience, diluted 1:250), hamster anti-PDPN (53–5381, eBioscience, 1:100), rat anti-PTPRC (553080, BD Biosciences, 1:400).

### EdU labeling in vivo

Mice were injected s.c. with EdU (50 µg/g mouse) 17 hr prior to sacrificing the animals. Lungs were removed and cells were isolated and surface-stained as described in the corresponding section. For additional EdU-labeling cells were fixed, permeabilized and stained using the Click-iT Plus Alexa

Fluor 555 Picolyl Azide Toolkit (ThermoFisher Scientific, Germany) according to the manufacturer's instructions.

## Immunofluorescence staining

Organ cryosections were fixed in ice-cold methanol for 10 min at −20°C. Blocking was performed with 10% goat serum (ThermoFisher Scientific, Germany) for 1 hr at RT. The vasculature was stained for CD31 and proliferating cells for KI67 overnight at 4°C. The sections were subsequently incubated with the appropriate secondary antibody for 1 hr at RT. Pictures were taken using the Zeiss Cell Observer and image analysis was accomplished with Fiji.

The following primary antibodies were used: rat anti-CD31 (553370, BD Biosciences, 1:50), rabbit anti-KI67 (GTX 16667, Gene Tex, 1:100), goat IgG (A11006, ThermoFisher Scientific, 1:500), goat IgG (A11071, ThermoFisher Scientific, 1:500).

## Cell culture

HUVECs were purchased from PromoCell (Heidelberg, Germany) and cultured in Endopan 3 with 3% FCS and supplements (PAN Biotech, Aidenbach, Germany) at 37°C, 5% $CO_2$ and high humidity. Passages < P6 were used for all experiments and cells were tested negative for mycoplasma contamination. For Lentiviral transduction (pLenti-based overexpression or pGIPZ-based silencing of certain molecules), $1 \times 10^5$ cells were seeded and 24 hr later lentivirus was added for 16 hr. Media was changed to Endopan with supplements for 24 hr before starting the selection with puromycin (P; 0.4 µg/ml) or neomycin (N; 200 µg/ml) or both (P 0.3 µg/ml and N 150 µg/ml) for 3–4 days.

The following lentiviral vectors were used: pLenti-CMV-EV-puromycin[Res] (control), pLenti-CMV-EV-neomycin[Res] (control), pLenti-CMV-SMAD6-puromycin[Res] (overexpression), pLenti-CMV-SMAD7-neomycin[Res] (overexpression), pGIPZ-ns-shRNA (control), pGIPZ-shSMAD6-118 (depletion), pGIPZ-shSMAD7-115 (depletion).

Cells were starved in Endopan without supplements for 2–4 hr before treating with 5 ng/ml BMP9 (R and D Systems, Minneapolis, Minnesota, US) or 10 ng/ml TGFB (R and D Systems, Minneapolis, Minnesota, US) for 30 min. For EdU incorporation analysis EdU was directly added to the subconfluent cells at a final concentration of 10 µM for 4 hr. Harvesting, fixation, permeabilization and staining were performed using the Click-iT EdU Flow Cytometry Assay Kit Alexa Fluor 647 (ThermoFisher Scientific, Germany) according to the manufacturer's protocol. Cells were analyzed on a BD FACSCanto II (BD Bioscience, Heidelberg, Germany). MTT assay was performed in 96well plates according to manufacturer's instructions (Roche, Germany). Migration analysis was performed by scratch assay in 24well plates. To prevent cell proliferation during migration analysis cells were treated with 10 µg/ml mitomycin C (Sigma, Munich, Germany) for 1.5 hr prior to scratching the monolayer. Pictures were taken either manually at certain time points (as indicated in the figures) at an Olympus IX71 Microscope (10x) or automatically every hour at an Olympus CellR Microscope with cell incubation chamber (10x). Image analysis was accomplished with Fiji.

## Elisa

Ligand level in tissue lysates were determined by ELISA according to the manufacture's protocols. BMP6 was measured by the ELISA Kit for Bone Morphogenetic Protein 6 (BMP6) (SEA646Mu, Cloud Clone, Katy, Texas, US), BMP9 was measured by the Mouse BMP9 ELISA Kit (ELM-BMP9, RayBiotech Norcross, Georgia, US) and TGFB1 was measured by the TGF-β1 Quantikine ELISA Kit (MB100B, R and D systems, Minneapolis, Minnesota, US). For BMP6 and BMP9 determination in lung and brain 100 ng/µl total protein concentration, for BMP6 and BMP9 measurement in heart 10 ng/µl total protein concentration and for TGFB1 detection in lung and heart 500 ng/µl total protein concentration was utilized.

## Immunoblot

Cells (whole tissue fragments or HUVEC) were lysed in RIPA buffer (1% NP-40, 0.1% Sodium dodecyl sulphate, 0.5% Sodium deoxycholate, 10% glycerol, 5 mM EDTA) supplemented with proteinase inhibitor mix (Serva Electrophoresis, Heidelberg, Germany) and sodium orthovanadate (Sigma, Munich, Germany). Protein lysates were separated on 10% polyacrylamide-SDS gels and blotted on nitrocellulose membranes. Membranes were blocked with 3% BSA and incubated with the indicated

primary antibodies at 4°C over night. Horseradish peroxidase-conjugated secondary antibodies were used for chemiluminescence detection. For detection, either an AGFA classic EOS developer (AGFA, Mortsel, Belgium) or an Amersham Imager 600 (GE healthcare, Little Chalfont, UK) was used.

The following primary antibodies were used: rabbit anti-pSMAD1/5/8 (D6656/Vli31, Maine Medical Center, 1:2000), rabbit anti-SMAD1 (9743S, Cell Signaling, 1:1000), rabbit anti-pSMAD2/3 (D6658, Maine Medical Center, 1:2000), rabbit anti-SMAD2/3 (5678, Cell Signaling, 1:1000), rabbit anti-ACTB (sc-1616, Santa Cruz Biotechnology, 1:5000), goat IgG (P0448, Dako, 1:5000).

## RNA isolation and quantitative PCR (qPCR) analysis

Total RNA of FACS-sorted mouse ECs was isolated with Arcturus PicoPure RNA Isolation Kit (ThermoFisher Scientific, Germany) and RNA of HUVECs was isolated with RNeasy Mini Kit (Qiagen, Hilden, Germany) according to the manufacturer's protocols. cDNA was synthesized with QuantiTect Reverse Transcription Kit (Qiagen, Hilden, Germany) according to the manufacturer's instructions. Subsequent qPCR was performed with TaqMan gene expression assay (ThermoFisher Scientific, Germany), TaqMan Fast Advanced Mastermix (ThermoFisher Scientific, Germany) and the StepOnePlus Real-Time PCR System (ThermoFisher Scientific, Germany). *Actb* (mouse EC) or *HPRT* (HUVEC) was used for data normalization.

The following TaqMan gene expression assays (ThermoFisher Scientific, Germany) were used: Acta2 (Mm00725412_s1), Actb (Mm00607939_s1), Bmpr2 (Mm00432134_m1), Ccnb1 (Mm03053893_gH), Ccnb2 (Mm01171453_m1), Cdk1 (Mm00772472_m1), Cyr61 (Mm00487501_g1), Fgfr1 (Mm00438930_m1), HPRT (Hs02800695_m1), Icam1 (Mm00516023_m1), Kdr (Mm01222421_m1), Notch3 (Mm01345646_m1), Nr2f2 (Mm00772789_m1), Ptprc (Mm01293577_m1), Sema3c (Mm00443121_m1), Smad6 (Mm01171378_m1), SMAD6 (Hs00178579_m1), Smad7 (Mm00484742_m1), SMAD7 (Hs00998193_m1), Stat1 (Mm00439531_m1), Tgfbr2 (Mm00436977_m1).

## RNA-sequencing (RNA-seq)

Total RNA from mouse lung EC was isolated using Arcturus PicoPure RNA Isolation Kit (ThermoFisher Scientific, Germany) according to the manufacturer's instructions. DNA was removed by treating with RNase-free DNase Set (Qiagen, Hilden, Germany). Quality control was performed by Qubit (ThermoFisher Scientific, Germany) and Bioanalyzer (Agilent, Waldbronn, Germany) measurements. Sequencing library was generated with 10 ng of total RNA using the SMARTer Ultra Low RNA Kit for Illumina Sequencing (Clontech, Mountain View, California, US) according to manufacturer's protocol. Sequencing reads (100 bp Paired-End) were generated on the HiSeq2000 platform (Illumina, San Diego, California, US) with four samples per lane.

## RNA-seq data processing and analysis

The sequenced reads were aligned to the mouse reference genome mm10 using STAR (*Dobin et al., 2013*) allowing up to 10 mismatches. The average transcriptome coverage was $37.8 \pm 4$. DEseq2 was used to test for differential gene expression (mm10, RefSeq gene annotation) (*Love et al., 2014*). RNA-seq data are available at the NCBI Gene Expression Omnibus (GEO) under accession number GSE86600. Only transcripts with an RPKM $\geq 1$ in at least one sample were considered for further analysis. Significantly differentially expressed genes were defined as more than 1.75-fold regulated (p<0.05). Hierarchical clustering was performed using GenePattern Software (*Reich et al., 2006*). For functional annotation with Reactome gene sets, the Molecular Signature Database (MSigDB) (*Subramanian et al., 2005*) was used. Upstream analysis with activation state prediction was generated by using QIAGEN's Ingenuity Pathway Analysis (IPA, QIAGEN Redwood City). According to the manual, "the upstream regulator analysis is based on prior knowledge of expected effects between transcriptional regulators and their target genes stored in the Ingenuity Knowledge Base. The analysis examines how many known targets of each transcription regulator are present in the user's dataset, and also compares their direction of change (i.e. expression in the experimental sample(s) relative to control) to what is expected from the literature in order to predict likely relevant transcriptional regulators. If the observed direction of change is mostly consistent with a particular activation state of the transcriptional regulator ('activated' or 'inhibited'), then a

prediction is made about that activation state.' The list of chromatin modifying enzymes has been described previously (*Plass et al., 2013*).

## Tagmentation-based whole genome bisulfite sequencing (T-WGBS) and validation by quantitative mass spectrometry (MassARRAY)

Genomic DNA from sorted EC was isolated using the QIAamp DNA Micro Kit (Qiagen, Hilden, Germany) according to the manufacturer's instructions. T-WGBS was essentially performed as described previously (*Wang et al., 2013*) using 30 ng genomic mouse DNA as input. Three sequencing libraries were generated per sample and each was sequenced 100 bp, paired-end on one lane of a HiSeq2000 (Illumina, San Diego, California, US) machine.

Validation of selected T-WGBS data was performed by MassARRAY (Agena Bioscience, Hamburg, Germany) as described previously (*Sonnet et al., 2014*) using primers listed in the *Supplementary file 1*. Forward primers for MassARRAY were 5' tagged with AGGAAGAGAG, reverse primers with CAGTAATACGACTCACTATAGGGAGAAGGCT.

## T-WGBS: Sequence alignment and cytosine methylation estimation

We used a mapping method as described earlier (*Johnson et al., 2012*) modified for reads from tagmentation-based whole-genome bisulfite sequencing. Briefly, the mm10 reference genome was transformed in silico for both the top strand (C to T) and bottom strand (G to A). Before alignment, adaptor sequences were trimmed using SeqPrep (*St John, 2016*; https://github.com/jstjohn/Seq-Prep). Then the first read in each read pair was C-to-T converted and the 2nd read in the pair was G-to-A converted. The converted reads were aligned to a combined reference of the transformed top (C to T) and bottom (G to A) strands using BWA (bwa-0.6.1-tpx) (*Li and Durbin, 2009*) with default parameters except the quality threshold for read trimming (-q) of 20 and the Smith-Waterman for the unmapped mate disabled (-s). After alignment, reads were converted back to the original states and reads mapped to the antisense strand of the respective reference were removed. Duplicate reads were removed using Picard MarkDuplicates (http://picard.sourceforge.net/). Reads with alignment scores less than one were filtered before subsequent analysis. Total genome coverage was calculated using the total number of bases aligned from uniquely mapped reads over the total number of mappable bases in the genome.

At each cytosine position, reads that maintain the cytosine status were considered methylated, and the reads that have cytosine converted to thymine were considered unmethylated. Only bases with Phred-scaled quality score of ≥20 were considered. For libraries prepared with the tagmentation protocol, first 9 bp of the second read and last 9 bp before the adapter in the first read were excluded from methylation calling. Bisulfite conversion rates were estimated using the methylation level at CH sites.

## T-WGBS: DMR calling and annotation

DMR calling was performed as described previously (*Bauer et al., 2016*) with minor modifications. In brief, DMR calling was conducted on methylation data without smoothing (due to sufficiently high coverage) with the bsseq v1.60 package (*Hansen et al., 2012*) for R statistical software v3.2.2. We applied a coverage filter that required a minimum coverage of 8x per CpG in at least two of the three samples per group. The DMR model comprised at least three consecutive significant CpGs (each p<0.05, t-test) and a minimum difference of 10% in mean methylation between groups.

Intersection with known regulatory genomic features was performed with Galaxy (*Afgan et al., 2016*) using published datasets (H3K4me3 ChIPseq: ENCSR000CAR; H3K4me1 ChIPseq: ENCSR000-CAQ; DNase Hypersensitivity: ENCSR000CNM) and enhancer-promoter units (EPU) defined in mouse lung tissue (*Shen et al., 2012*; *Yue et al., 2014*). For data visualization tracks were loaded into the IGV browser (*Robinson et al., 2011*). Motif searches for known transcription factor binding sites (TFBS) in DMRs and TFBS enrichment over background were analyzed with the 'findMotifsGenome.pl' script of the HOMER tools software package (*Heinz et al., 2010*).

Genomic annotation of DMRs to the nearest TSS was obtained with the 'annotatePeaks.pl'-script of the HOMER tools software package (*Heinz et al., 2010*) to genome version mm10. Functional annotation of up to three nearby genes of DMRs was accomplished using the GREAT tool (*McLean et al., 2010*). For enrichment analysis among a ranked list of genes (sorted according to

the number of DMRs or rel. gene regulation) the Gene Set Enrichment Analysis (GSEA) Software was applied (*Subramanian et al., 2005*). Tagmentation-based WGBS data are available at the NCBI Gene Expression Omnibus (GEO) under accession number GSE87374.

## Interactome analysis

To gain further functional insights into the mechanisms of EC quiescence, we generated a network from genes that were significantly differentially regulated (p<0.05). For generating such a network, we used the DIAMOnD algorithm (*Ghiassian et al., 2015*), which creates a network starting from a set of proteins (so-called seed set) by integrating related proteins into the network based on their topological relevance to the initial seed set. DIAMOnD first ranks all proteins in the interactome according their connectivity significance with respect to the seed set. Next, the protein with the highest rank, that is, lowest p-value, is added to the initial network. The procedure is repeated with the extended seed set, until the incorporation of seed proteins saturates.

Genes with a FC above 1.75 or below −1.75, were mapped to their corresponding gene products and selected as seeds for generating the network, with a total of 972 genes being up and 2,251 genes being down regulated (a total of 3,223 regulated genes). The mouse interactome comprised 7,267 proteins and 18,382 protein interactions, obtained by merging publicly available data from the major public protein-interaction databases. In the end, 227 (23.4%) upregulated and 618 (27.5%) downregulated proteins were covered in the assembled mouse interactome and used as seed proteins to generate the EC specific interaction network.

Given the incompleteness of the mouse interactome, a large number of DIAMOnD iterations were necessary to connect all seed proteins through interactions, that is, 5,943 iterations were needed to link 99.3% of the proteins. In consequence, the resulting network was very large, covering about 93% of the mouse interactome. Since we were primarily interested in proteins functionally related to EC quiescence, we applied a strict threshold, terminating the network generation after the first steep increase in the proportion of incorporated seeds started to flatten (iteration: 265). The EC quiescence specific network with 745 proteins and 1,661 interactions is shown in the *Figure 1— figure supplement 2*. A total of 484 seed proteins (57.3%) were connected in the network through 261 linker proteins, which were of high functional interest with respect to EC quiescence.

Next, we assessed whether proteins forming the EC quiescence network were functionally coherent with respect to the initial seed set. To this end, we performed a functional enrichment with g: Profiler (*Reimand et al., 2016*) of two protein sets, namely (i) proteins of the seed set covered by the interactome, and (ii) proteins of the network, and compared the respective outcomes focusing on KEGG pathways. As background set, we used the 7,267 proteins comprising the mouse interactome. As the reduced seed set might yield a distinct enrichment, we performed in addition a functional enrichment for proteins of the complete initial seed set. The complete list of enriched pathways for each protein is shown in *Figure 1—source data 1*. In general, we observed a large overlap of enriched pathways between the seeds and the EC quiescence network. These findings emphasized that proteins captured in the network were of functional relevance with respect to proteins regulated in EC quiescence.

To obtain more insights from the generated network, we performed network clustering identifying densely connected subgraphs, so called communities. The 23 detected cluster were ranging in size from two to 165 proteins. For each of the cluster, we performed functional enrichment analysis using g:Profiler (*Reimand et al., 2016*).

## Acknowledgements

We would like to acknowledge the excellent technical support of Stella Hertel and Benjamin Schieb (H.G.A. laboratory) as well as Oliver Mücke, Monika Helf and Marion Bähr (C.P. laboratory). We thank the Flow Cytometry Core Facility and the Central Animal Laboratory, German Cancer Research Center (DKFZ), for providing excellent services. We thank the High Throughput Sequencing unit of the Genomics and Proteomics Core Facility (DKFZ) for providing excellent library preparation, sequencing and QC analysis services (RNA-seq QC and alignment by Christopher Previti). We thank the Light Microscopy Core Facility (DKFZ) for providing instruments and excellent macro generation by Damir Krunic. We thank Lidia Mateo (IRB Barcelona) for help with the EC maturation network representation.

## Additional information

### Funding

| Funder | Grant reference number | Author |
|---|---|---|
| Bundesministerium für Bildung und Forschung | e:Med program for systems biology (PANC-STRAT consortium grant no. 01ZX1305) | Tobias Bauer |
| Bundesministerium für Bildung und Forschung | DZL and DZHK | Christoph Plass |
| Deutsche Forschungsgemeinschaft | SFB1324 | Hellmut Augustin |
| Deutsche Forschungsgemeinschaft | SFB873 | Hellmut Augustin |
| Deutsche Forschungsgemeinschaft | SFB-TR-23 | Hellmut Augustin |
| European Commission | SyStemAge | Hellmut Augustin |

The funders had no role in study design, data collection and interpretation, or the decision to submit the work for publication.

### Author contributions

Katharina Schlereth, Conceptualization, Data curation, Formal analysis, Supervision, Validation, Investigation, Visualization, Methodology, Writing—original draft, Writing—review and editing; Dieter Weichenhan, Daniel Lipka, Methodology, Writing—review and editing; Tobias Bauer, Data curation, Software, Formal analysis, Writing—review and editing; Tina Heumann, Validation, Investigation, Methodology, Writing—review and editing; Evangelia Giannakouri, Investigation, Writing—review and editing; Samira Jaeger, Formal analysis, Methodology, Writing—review and editing; Matthias Schlesner, Patrick Aloy, Supervision, Writing—review and editing; Roland Eils, Conceptualization, Supervision, Writing—review and editing; Christoph Plass, Conceptualization, Resources, Supervision, Funding acquisition, Writing—original draft, Project administration, Writing—review and editing; Hellmut G Augustin, Conceptualization, Resources, Data curation, Supervision, Funding acquisition, Validation, Writing—original draft, Project administration, Writing—review and editing

### Author ORCIDs

Tobias Bauer http://orcid.org/0000-0002-4961-3639
Daniel Lipka http://orcid.org/0000-0001-5081-7869
Matthias Schlesner https://orcid.org/0000-0002-5896-4086
Hellmut G Augustin http://orcid.org/0000-0002-7173-4242

### Ethics

Animal experimentation: All mice were handled according to the guidance of the Institute and as approved by the German Cancer Research Center (No. DKFZ305).

### Decision letter and Author response

Decision letter https://doi.org/10.7554/eLife.34423.030
Author response https://doi.org/10.7554/eLife.34423.031

## Additional files

### Supplementary files

• Supplementary file 1. Key resource table.
DOI: https://doi.org/10.7554/eLife.34423.017

• Transparent reporting form

DOI: https://doi.org/10.7554/eLife.34423.018

## Major datasets

The following datasets were generated:

| Author(s) | Year | Dataset title | Dataset URL | Database, license, and accessibility information |
|---|---|---|---|---|
| Schlereth K, Wei-chenhan D, Bauer T, Eils R, Augustin HG, Plass C | 2016 | DNA methylation profiling of vascular maturation | https://www.ncbi.nlm. nih.gov/geo/query/acc. cgi?acc=GSE87374 | Publicly available at the NCBI Gene Expression Omnibus (accession no: GSE87374). |
| Schlereth K, Previti C, Augustin HG | 2016 | Expression profiling of infant and young adult lung endothelial cells by RNA-sequencing | https://www.ncbi.nlm. nih.gov/geo/query/acc. cgi?acc=GSE86600 | Publicly available at the NCBI Gene Expression Omnibus (accession no: GSE86600). |

The following previously published datasets were used:

Shen Y, Yue F, McCleary DF, Ye Z, Edsall L, Kuan S, Wagner U, Dixon J, Lee L, Lobanenkov VV, Ren B2012A map of the cis-regulatory sequences in the mouse genomehttps://www.ncbi.nlm.nih.gov/geo/query/acc. cgi?acc=GSE29184Publicly available at the NCBI Gene Expression Omnibus (accession no: GSE29184). The Mouse ENCODE Consortium2014A comparative encyclopedia of DNA elements in the mouse genomehttp:// mouse.encodedcc.org/Publicly available at the Mouse ENCODE Consortium website

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
