## [Decision Letter]

[Editors’ note: a previous version of this study was rejected after peer review, but the authors submitted for reconsideration. The first decision letter after peer review is shown below.]

Thank you for submitting your work entitled "Transcriptional and epigenetic programming during the transition from developmental angiogenesis to vascular quiescence" for consideration by *eLife*. Your article has been reviewed by three peer reviewers, one of whom is a member of our Board of Reviewing Editors and the evaluation has been overseen by a Senior Editor. The reviewers have opted to remain anonymous.

Our decision has been reached after consultation between the reviewers. Based on these discussions and the individual reviews below, we regret to inform you that your work will not be considered further for publication in *eLife*.

Whilst the reviewers found your work of interest they felt it lacked sufficient experimental validation of several claims. The identification of genetic and epigenetic regulation mechanisms in vessel maturation and quiescence is indeed seen as a timely and important question, and the reviewers see that your data has the potential to become an important catalogue for future studies. However, in the absence of strong validation of function, in particular in vivo and given that it remains very unclear what aspect of the regulation truly relates to quiescence rather than organ specific vessel adaptations, it was felt there will need to be a substantial amount of extra work to clarify the true value of the data. The importance of Tgfb in vessel maturation per se is not deemed novel, and it is felt your work adds little new insight to advance the understanding of how this pathway achieves maturation. The epigenetic regulation, the methylome analysis and the identification of hot spots of DMRs in is seen as a potential strength, but also lacks functional validation. As *ELife* does not encourage asking for extensive revisions that will likely take a lot of time. we decided it will be most appropriate to return the manuscript to you. If, however you feel you can address the full critique below and can identify a set of key experiments that will validate the claims within a short time, you may of course appeal our decision.

Reviewer #1:

The manuscript by Schlereth and colleagues addresses the important question of how endothelial cell maturation and quiescence is regulated through genetic and epigenetic mechanisms. The authors take an unbiased whole genome approach using RNA seq to study gene expression, combined with analysis of gain and loss of methylation to assess epigenetic regulation. The authors use mouse lung as model selecting postnatal day 8 as adolescent stage, and 8-12 weeks as adult stage.

FACS sorted cells are processed for analysis. Comprehensive bioinformatic analysis identified regulated gene networks associated with quiescence and qPCR of a few genes confirms the RNA seq data. Probably the most interest new data relate to the identification of prominent differential methylation of intragenic clusters in ECs during methylation, with intronic enhancers prominently featuring as differentially methylated regions. Key regulated pathways include JAK-STAT and TGFb signaling, in the case of the latter, strong upregulation of the inhibitory smads 6 and 7. These are also targets of epigenetic regulation and could be confirmed in isolated cells from brain and heart.

Overall, the authors conclude that the identified regulatory mechanisms drive endothelial quiescence and maturation.

Whilst the comprehensive approach is interesting and contains valuable information, there are aspects that reduce enthusiasm as the conclusions appear somewhat overstated or unclear. Given that the study is essentially based on sorted cells from one organ, it is unclear whether the gene networks identified truly signify general mechanisms of quiescence or rather relate to maturation of the lung vasculature in particular. For example, it seems possible that the lung vasculature is particular active in immune-related signaling given the strong exposure to pathogens. It is noteworthy that the authors do not state what the pathogen status of the animals is. They were purchase from Taconic, but no other information is provided. The regulation of angiotensin is also interesting as the lung vasculature runs under different pressure regimes compared to the rest of the body. Again, is this lung specific? Some genes that are found would further suggest that not all sequences come from endothelial cells. For example, Notch3 is normally not expressed by EC to my knowledge. The Results sections does not contain information how the authors controlled for purity. The Materials and methods section however show that the isolation is not only done by FACS but preceded by dynabead magnetic sorting. Given that the lung tissue develops substantially between the selected stages, it would be important to understand whether the two stages had similar levels of non-EC contamination. The isolation and digestion procedures are bound to be affected differently. This needs to be explained.

As the most strongly regulated pathway, the authors identify Tgfb, in particular smad6 and smad7. To my understanding, these are activated differentially downstream of BMP and Tgfb signaling. Although they act as repressors, their engagement in a feedback loop would normally imply that Tgfb signaling may be activated in maturation. There is little detailed analysis of this question. A general inhibition of either Tgfb or BMP would be expected to drive endothelial activation, not necessarily quiescence. A detailed analysis of the phosphorylation status of R-smads in vivo would seem important to understand this.

Overexpression of just the inhibitory smads may not at all reproduce the situation the authors find in vivo.

In conclusion, the present work provides an interesting starting point, with data that may become valuable for the field with time, once further functional validation has been done. It would be unreasonable to ask for full validation of the extensive regulatory networks. But it may be possible for the authors to further validate the role of Tgfb pathway, in particular to dissect whether the assumed quiescence regulation is indeed only achieved through epigenetic regulation of the inhibitory smads, or not (ligand levels etc.). If so, it would be critical to investigate whether such a mechanism provides endothelial cells with a state of resistance to ligand induced activation? Can the authors modify the methylation experimentally?

If no further validation/clarification can be provided, the present work will be a catalogue plus some conceptual advance that however remains to be tested.

I am unsure whether the authors will be able to provide sufficient confirmation with a limited revision for *eLife*.

Reviewer #2:

This study provides a description of the transcriptomic and epigenetic landscape of endothelial cell maturation by performing analyses on ECs isolated from day 8 and young adult mice (8-10 weeks old). Numerous changes are cataloged. Robust changes are seen in Smad6 and Smad7 and their roles as anti-angiogenic, anti-proliferative signaling molecules is validated by in vitro studies.

A detailed analysis of Smad6 and 7 on control of EC responses to BMP and lateral branching was published last year by Bautch's group. It is not cited here.

In the Discussion section it is stated that "SMAD6 and SMAD7 have originally been identified as vascular SMADs and are, in fact, expressed almost exclusively in EC (Topper et a., 1997)" citing a 20 year old paper. I doubt that this is true. A PubMed search suggests that they are ubiquitously expressed.

Reviewer #3:

The manuscript seeks to investigate the transcriptional and epigenetic mechanisms that control endothelial quiescence. This is still a fairly unexplored area of vascular biology; thus, the study is timely and welcome. To achieve their aim, the Authors use FACS sorted lung EC from newborn mice (P 8) and young adult mice (8-12 weeks) for RNA Seq and methylation analysis.

The study takes a novel approach to the question of quiescence. Transcription profiling identifies over 2000 genes differentially expressed in newborn vs young adult; GO analysis identifies genes involved in multiple families including cell cycle, and proliferation analysis of EC from various organs using EdU and Ki567 profiling shows decreased EC proliferation in the adult. Figure Figure 1 shows two of the pathways that may be involved in this phenotypic switch, namely the JAK-STAT pathway and the TGFb pathway. Given that the focus of the study is on quiescence, it would be important to provide the list of genes involved in cell cycle regulation. The network analysis in Figure 1—figure supplement 2 is not very helpful; it could be removed and replaced by small selected networks with each gene labelled.

Of the partial list of genes involved in angiogenesis shown in Figure 2, the very noticeable absents are members of the VEGF/VEGFR families. The Authors acknowledge this, but do not discuss the possible reasons. This should be addressed in the Discussion section. Also, this somewhat weakens the argument made throughout the paper that angiogenic pathways are regulated during the progression to adulthood. A key player in this process appears to be TGFb, which becomes the focus of the second half of the manuscript. Therefore, I would recommend altering the sections of the paper where they claim that angiogenic pathways are crucially regulated. The title also should be modified, since it creates an expectation of a major focus on angiogenesis pathways, when in fact most of the validated analysis is on TGFb pathways.

Having profiled the transcriptome, the Authors proceed to profile the methylome. This is an interesting and novel experiment, and the result of potential value; other studies on EC methylation have been carried out but not on this model of young vs adult mouse EC. Data description and representation should provide more detail in the text and not only in the figure legends and Materials and methods section. Figure 3A shows a difference on CpG methylation and DMRs between newborn and young adult EC: is the difference statistically significant? Analysis of the location of DMRs identifies regions distal from the TSS as enriched in DMRs; the figure also shows EPU: enhancer promoterUnits, (defined in the figure legend not in the text): how are these identified?

Analysis of TF motifs within DMRs identifies FOX and ETS as the most represented and conclude that FOX motifs are demethylated in adult whilst ETS motifs acquire methylation in adult EC, suggesting that ETS pathways, active in newborn mice, are switched off in the adult, and the opposite for FOX. Based on these findings, the Authors conclude that there is a switch between FOX and ETS pathways which controls the progression to adult quiescent EC. This is an interesting hypothesis, but it is not demonstrated and therefore the conclusions are too speculative. Firstly, the ERG (ETS) binding site is significant in both gain and loss regions – this is not mentioned in the text. Also, levels of TF from these families do not support the "switch" model, since they find no difference in the FOX group, an increase in 3/5 IRF TF and a mix of up-regulation, down-regulation or no effect in the 5 ETS factors investigated. Thus, motif analysis is not consistent with TF expression; that may not necessarily be a conflict, since there are more factors in each family, and their expression levels may not be the key determinant of function. However, lack of correlation between TF expression and TF motif methylation somewhat weakens the theory of a "switch" and suggests more of a modulation. There is also an issue with methylation status, which on a gene to gene basis does not consistently predict induction or repression of the gene (Figure 5D and Figure 6D). Also, are any FOXO or ETS target genes differentially regulated between infant and young adult mice (c.f. Figure 2A)? this is not mentioned, and it seems an odd omission to the study. Crucially, validation of the role of this FOX – ETS switch needs to be carried out experimentally, by testing whether the changes in DMRs at these motifs are transcriptionally meaningful and result in differential gene expression of the TF targets, as well as differences in EC proliferation.

The study then moves on to the TGFβ pathway and SMADs signalling, with some straightforward in vitro validation of selected targets. Surprisingly, the TF motif analysis there is no sign of SMAD motifs. This is not discussed – it would be interesting to know the Authors view on this. Involvement of TGFb in maturation/differentiation is not a particularly novel finding, but it is still novel in this model and with this approach.

There are two separate stories in this paper, which at present do not sit comfortably together: a more preliminary one, to do with the FOX/ETS switch and the other one, more mature, on the TGFb pathway. In summary, this is a very interesting study with novel data of potential interest to the vascular biology community. However, some of the conclusions are not supported by the data and require more validation.

[Editors’ note: what now follows is the decision letter after the authors submitted for further consideration.]

Thank you for submitting your article "Genome-wide profiling of endothelial cells identifies TGFß family signaling as key regulator of vascular quiescence" for consideration by *eLife*. Your article has been reviewed by three peer reviewers, and the evaluation has been overseen by Didier Stainier as the Reviewing and Senior Editor. The following individuals involved in review of your submission have agreed to reveal their identity: Gou Young Koh (Reviewer #1); Rosemary Akhurst (Reviewer #3).

The reviewers have discussed the reviews with one another and I have drafted this decision to help you prepare a revised submission.

None of the reviewers' queries involve additional experiments, except possibly one concerning Smad6 and 7 (see below). Also, in order for you to see the full extent of the reviews, I have kept them separate. Hopefully, you will find them helpful as you revise the manuscript.

Summary:

The aim of this revised manuscript from Schlereth et al., manuscript is to discover which molecules drive endothelial cell (EC) quiescence. To achieve this, they compare EC gene expression changes that occur as new born mice progress to adulthood and establish and maintain a quiescent adult endothelium. The group undertake an "-omics" comparison of FACS sorted (CD34+CD31+) lung ECs harvested from suckling mice (8-10dpp) versus those harvested from adult mice (6-8 weeks dpp) mice, including transcriptomic profiling and DNA methylomics.

One question that came up in the discussion was in regard to the lack of Smad6 and 7 Western blots (or in situ or immunostaining in vivo) to look at the level of expression of these genes/proteins. Was it attempted and not successful? Two reviewers thought that it would be a very nice addition to the paper.

Reviewer #1:

The manuscript by Schlereth et al., tackle the important yet unanswered question in the field of vascular biology, the acquisition of quiescence in endothelial cells (ECs). The authors compared freshly isolated ECs from infant and young adult mouse lungs to gain insight about the dominant regulators that generate stable vasculature in healthy, normal mice. By incorporating unbiased whole genome approach with transcriptome and methylome analysis, and also by analyzing ECs from multiple organs, this study provides valuable insights and information which has broad applicability for future studies in the field. The convergence of evidence from transcriptome analysis and methylome analysis all pointing toward TGFβ signaling and in vitro validation of the role played by Smad6 and Smad7 on EC quiescence make a strong case for TGFβ being a key regulator of EC quiescence. Few minor adjustments would help improve this manuscript before publication.

1) Subsection “TGFs family signaling is inhibited in quiescent EC”: "Surprisingly, members of the VEGF/VEGFR pathway were not identified as being transcriptionally regulated." This sentence is rather confusing, given that in Figure 2A, heatmap showed upregulation of Vegfa in EC of young adult, and also in Figure 1—figure supplement 1B, EC of young adult seems to have higher expression of Kdr (Vegfr2) compared to EC of infant. Does it mean that heatmap and RT-PCR results are not statistically significant? Or does it mean that IPA analysis of upstream regulators did not reveal VEGF/VEGFR pathway as one of significantly altered signaling pathways? Also, in the Discussion section, the authors wrote "Since the expression of the primary VEGF signal transducing receptor Kdr/Vegfr2 was not changed,…". Is the expression level of Kdr not changed in RNA-seq or RT-PCR data, or both? More detailed explanation is required to avoid confusion.

2) Subsection “Clustered intragenic loss of methylation regulates TGFβ family and semaphorin signaling in quiescent endothelial cells”: "The correlation of quiescence-related expression and methylation profiles revealed an upregulated expression of genes with hypomethylated DMRs in adult EC independent of the genomic loci of one or several DMRs." This is clearly not the case for Sema6a as shown in Figure 4 and Figure 4—figure supplement 1A. Intragenic regions of Sema6a showed multiple DMRs, but its expression level is downregulated in young adult ECs. Smad7, on the other hand, showed multiple DMRs in intragenic regions and showed significant upregulation in expression level in young adult ECs, agreeing with the authors' claim. This discrepancy clearly suggests that hypomethylation does not guarantee upregulation of expression level of the affected genes. The authors should at least discuss this matter in more detail and avoid writing generalized statements concerning hypomethylation and the direction of gene expression changes, which seem oversimplified in the current version of the manuscript.

3) Does the number of DMRs in a given gene have a direct correlation with the extent of expression change of that gene? In the Discussion section, the authors wrote "The presence of clustered DMRs was even more likely associated with differential expression changes of the affected gene, suggesting that we have identified intronic regions with high regulatory potential during the acquisition of vascular quiescence." However, heatmap in Figure 4—figure supplement 1B showed strong expression changes of Tek, which has only one DMR. Several other genes also showed significant expression changes while having a small number of DMRs (Ltbp3, Nid1, Tgfbr2, etc.). If there is no direct correlation, what is the meaning of Smad6 and Smad7 containing the largest cluster of DMRs? Are those just a lucky coincidence or are there more profound biological meaning to it? More detailed explanation would definitely help readers to understand this manuscript better.

Reviewer #2:

The manuscript has significantly improved in terms of focus and clarity. There are a few aspects which in my view could be improved.

1) Several pathways are mentioned, and some are validated in the course of the study. It is not always clear how these were prioritised. Early on, they state that "Functional enrichment analysis of the EC quiescence interactome led to the identification of JAK‐STAT signaling being highly enriched". Figure 1—source data 1 lists several other pathways, as expected. The text (subsection “Endothelial transcriptomic changes during acquisition of vascular quiescence”) should be reworded to make it clearer that several pathways were identified (some of which somewhat surprising to this reviewer), and that the JAK-STAT was the most significant. Strangely, this pathway is not mentioned again and is not validated. Is there a reason for this?

2) Figure 1—figure supplement 2 is not particularly useful, given that no nodes have been assigned labels (given the size of the network, that would be impossible). It is not easy to find a reader-friendly way to present this kind of analysis. It might be more helpful to select a few networks, including the Jak-Stat, Notch and TGF b pathways, and show network analysis related to these in detail, with names and direction of change in infant vs adult EC.

3) The Authors move on to a selection of angiogenic molecules (Figure 2). How were these genes selected? Was the list manually curated or did they use DAVID or another GO software to select angiogenic regulators?

4) They eventually focus on TGFb. In subsection “TGFs family signaling is inhibited in quiescent EC” they state: "upstream analysis of the complete set of differentially expressed genes including activation state prediction revealed TGFß1 as the most significantly inhibited growth factor". How does this relate to the pathways shown in Figure 1—source data 1? the process which led them to focus on this pathway, in the context of other pathways identified here, should be more clearly explained

5) How were DMR assigned to genes? Subsection “Clustered intragenic loss of methylation regulates TGFβ family and semaphorin signaling in quiescent endothelial cells”: "As enhancer regions could influence the expression of distant genes, up to three nearby genes were assigned to the dependent DMRs to identify genes likely affected by differential methylation". Please provide reference related to assignation method.

6) The correlation of quiescence-related expression and methylation profiles is an important aspect of this study, since it confirms the hypothesis that the cells' phenotype, and thence the infant to adult transition, is controlled by DNA methylation. Therefore, some of the data in Figure 4—figure supplement 1 should be in the main paper and should be described in more detail. There is a lot of information in this figure, which should be presented and not just briefly described in the Figure legend. Also, Subsection “Clustered intragenic loss of methylation regulates TGFβ family and semaphorin signaling in quiescent endothelial cells”: "these data suggest that vascular quiescence is accompanied by a prominent loss of methylation in intragenic DMR clusters preferentially affecting and regulating the expression of genes involved in the regulation of TGFβ family, semaphorin and NOTCH signaling in this process". Is there a GO analysis showing how these pathways rank compared to others? It is interesting to focus on these pathways, but it should be clear whether these are the top pathways or whether they have been selected because of their functional relevance.

7) EC differentiation and maturation, the acquisition of quiescence and their angiogenic potential are all aspects of overlapping processes; The Authors repeatedly use "angiogenesis" as the opposite of quiescence. The opposite of quiescence is proliferation, and indeed in the early part of the study the Authors measure the proliferation rate of the cells under investigation. But I don't think that it is correct to equate "proliferative" with "angiogenic", given that several stages of angiogenesis do not require proliferation.

8) The data on SMAD 6&7 overexpression (Figure 5, Figure 5—figure supplement 1 and Figure 5—figure supplement 2) is interpreted as confirming a key role for these two in the acquisition of endothelial quiescence. However, over-expression of SMAD6 alone does not affect proliferation, and the combined over-expression results in decreased cell viability. The interpretation of these results should be more cautious.

9) The conclusions are that the TGFb family is critical for the process. But what happened to the STAT pathway which was highlighted in Figure 1? This should be mentioned in the Discussion section.

Reviewer #3:

The aim of this revised manuscript from Schlereth et al. manuscript is to discover which molecules drive endothelial cell (EC) quiescence. To achieve this, they compare EC gene expression changes that occur as new born mice progress to adulthood and establish and maintain a quiescent adult endothelium. The group undertake an "-omics" comparison of FACS sorted (CD34+CD31+) lung ECs harvested from suckling mice (8-10dpp) versus those harvested from adult mice (6-8 weeks dpp) mice, including transcriptomic profiling and DNA methylomics. The authors should be congratulated on bioinformatics analysis and interpretation of a large body of genome wide expression data that is certainly correct and gives insights into the adult quiescent endothelial cell state. However, since e Life tends to be read by biologists, the text could be expanded slightly to make it easier for biologists with fewer quantitative skills to assess and follow.

The extensive genome-wide unbiased data reveals that components of active canonical TGF β and BMP signaling are down regulated in the adult quiescent state, exemplified by low levels of expression of receptors for TGF β and BMP and low levels of phosphorylated pSmad2/3 and pSmad1/5/8. This is interesting, and consistent with the widely accepted view of TGF β signaling serving in homeostatic regulation of epithelial and endothelial growth. The prediction would be that this signaling pathway would be upregulated when homeostasis is perturbed (e.g. by wounding, damage to endothelium, inflammation) in order to restore quiescence, a phenomenon that has been described extensively in epithelia and to a lesser extent for endothelia, but not addressed in the current study.

The major novel finding of the current study is that genes encoding the inhibitory SMADs, Smad6 and SMAD7, are released from the epigenetic suppression that exists in lung ECs of young mice, such that in the adult these genes show extensive DNA hypomethylation at intronic CpG clusters indicative of higher gene expression, and consistent with the observation of higher Smad6/7 transcript levels by RNAseq and TAQMan. Thus, a major difference between lung EC from suckling versus adult mice is that in the former Smad6/7 genes are epigenetically repressed, whereas in the adult excessive Smad6/7 expression suppresses active TGF β/BMP signaling. The authors also show that in brain and heart EC, there too is epigenetic suppression of Smad6/7 in suckling mouse lung EC and release from such suppression in adult lung EC, suggesting that this is a universal rather than tissue specific effect. The manuscript (a) provides a useful resource for individuals interested in analysis of EC gene expression, (b) helps to place into context earlier findings of contextual actions of TGF β on EC, and (c) forms the foundation for further mechanistic studies.

The previous reviewers asked for more in vivo validation of data. Although the authors go some way to provide this in Figure 5, the data in this figure is from total tissue lysate not from ECs, and would be include immune cells, epithelial cells and fibroblasts. It would be important to demonstrate that the high Smad6 and Smad7 transcript levels in adult versus young lung ECs are translated in to differential protein levels, either by Western and/or IHC.

If not already assessed in the literature or by the authors, future studies should address whether Smad6/7 protein is diminished in EC during pathological angiogenesis? NB Smad6 has at least as high expression in smooth muscle cells and pericytes, so it would be important to differentiate these two populations from EC.

---

## [Author Response]

[Editors’ note: the author responses to the first round of peer review follow.]

*Whilst the reviewers found your work of interest they felt it lacked sufficient experimental validation of several claims. The identification of genetic and epigenetic regulation mechanisms in vessel maturation and quiescence is indeed seen as a timely and important question, and the reviewers see that your data has the potential to become an important catalogue for future studies. However, in the absence of strong validation of function, in particular* in vivo *and given that it remains very unclear what aspect of the regulation truly relates to quiescence rather than organ specific vessel adaptations, it was felt there will need to be a substantial amount of extra work to clarify the true value of the data. The importance of Tgfb in vessel maturation per se is not deemed novel, and it is felt your work adds little new insight to advance the understanding of how this pathway achieves maturation. The epigenetic regulation, the methylome analysis and the identification of hot spots of DMRs in is seen as a potential strength, but also lacks functional validation. As ELife does not encourage asking for extensive revisions that will likely take a lot of time. we decided it will be most appropriate to return the manuscript to you. If, however you feel you can address the full critique below and can identify a set of key experiments that will validate the claims within a short time, you may of course appeal our decision.*

We sincerely appreciate the editor´s summarizing assessment of our manuscript. We would also like to thank the reviewers for their constructive and critical comments considering this work to have substantial potential for future studies on vascular maturation but at present lacking significant validation experiments. In a nutshell, the reviewers’ comments may be summarized as “exciting, but premature”. We essentially agree with this assessment. Admittedly, some of the data, particularly in the text of the Results section have not been spelled out in sufficient detail. Yet, when carefully going over the figures, it is clear that the answers to many of the reviewers’ comments are already in the manuscript (see response to specific comments below). From the epigenetics point of view, there is in the recent high-profile literature a lot of correlative work relating transcriptomic changes to epigenetic changes without substantial functional validation. We believe the present study stands out, because we have done such analysis in very close association to an important biological function employing freshly in vivo isolated cells. Based on this analysis, we have prototypically validated one of the most important identified pathways in cellular experiments in vitro and in expression profiling experiments in vivo. The starting point of our study (infant vs. adult EC) was amazingly simple. Yet, with all the vascular transcriptomic work pursued in recent years, this highly relevant biological differential has in no study been pursued – albeit being more than obvious. Along these lines, we do not think that the discovery of a key role of TGFs pathway signaling in vascular maturation is per se not novel. TGFs pathway functions during angiogenesis are through different receptors highly contextual and even many experts would likely be confused to answer if the net outcome of the pathway would in vivo be primarily pro- or antiangiogenic. We therefore consider the discovery of the pathway in the probably most relevant physiological vascular maturation process an important discovery. Likewise, it is obvious from our validation experiments in brain and heart (which has been extended in the revised manuscript) that the overriding process that we have studied is vascular quiescence and not organotypic lung EC differentiation.

As spelled out in detail in the response to the individual reviewers’ comments, we have addressed all comments through additional experiments and/or extensive editorial revisions. Importantly, the manuscript has been completely restructured. In doing so, we have taken up the very constructive comments of reviewer #3, who concluded that the original paper focused on two stories, the FOXO/ETS switch and the regulation of the TGFs pathway during vascular quiescence with both of them not being fully developed. We agree. In refocusing the revised manuscript, we removed the FOXO/ETS switch data (on which we will continue to work and will communicate them at a later stage separately when it is more mature), to focus on the systems biology transcriptional and methylome profiling, which has led to the discovery of the TGFs pathway as mechanism of physiological vascular quiescence. We point out towards this end the central role of signaling inhibition by epigenetically-regulated Smad6/7 expression. In line with the addressing of the additional reviewers’ specific comments, we believe that such revised structure gives the manuscript much more coherence and better readability. Along the same lines, as suggested by reviewers #1 and #3, the title of the manuscript has been rephrased to better fit with the content of the study towards: “Genome-wide profiling of endothelial cells identifies TGFß family signaling as key regulator of vascular quiescence”.

Reviewer #1:The manuscript by Schlereth and colleagues addresses the important question of how endothelial cell maturation and quiescence is regulated through genetic and epigenetic mechanisms. The authors take an unbiased whole genome approach using RNA seq to study gene expression, combined with analysis of gain and loss of methylation to assess epigenetic regulation. The authors use mouse lung as model selecting postnatal day 8 as adolescent stage, and 8-12 weeks as adult stage.FACS sorted cells are processed for analysis. Comprehensive bioinformatic analysis identified regulated gene networks associated with quiescence and qPCR of a few genes confirms the RNA seq data. Probably the most interest new data relate to the identification of prominent differential methylation of intragenic clusters in ECs during methylation, with intronic enhancers prominently featuring as differentially methylated regions. Key regulated pathways include JAK-STAT and TGFb signaling, in the case of the latter, strong upregulation of the inhibitory smads 6 and 7. These are also targets of epigenetic regulation and could be confirmed in isolated cells from brain and heart.Overall, the authors conclude that the identified regulatory mechanisms drive endothelial quiescence and maturation.Whilst the comprehensive approach is interesting and contains valuable information, there are aspects that reduce enthusiasm as the conclusions appear somewhat overstated or unclear. Given that the study is essentially based on sorted cells from one organ, it is unclear whether the gene networks identified truly signify general mechanisms of quiescence or rather relate to maturation of the lung vasculature in particular. For example, it seems possible that the lung vasculature is particular active in immune-related signaling given the strong exposure to pathogens. It is noteworthy that the authors do not state what the pathogen status of the animals is. They were purchase from Taconic, but no other information is provided. The regulation of angiotensin is also interesting as the lung vasculature runs under different pressure regimes compared to the rest of the body. Again, is this lung specific? Some genes that are found would further suggest that not all sequences come from endothelial cells. For example, Notch3 is normally not expressed by EC to my knowledge. The Results section does not contain information how the authors controlled for purity. The Materials and methods section however show that the isolation is not only done by FACS but preceded by dynabead magnetic sorting. Given that the lung tissue develops substantially between the selected stages, it would be important to understand whether the two stages had similar levels of non-EC contamination. The isolation and digestion procedures are bound to be affected differently. This needs to be explained.

*As the most strongly regulated pathway, the authors identify Tgfb, in particular smad6 and smad7. To my understanding, these are activated differentially downstream of BMP and Tgfb signaling. Although they act as repressors, their engagement in a feedback loop would normally imply that Tgfb signaling may be activated in maturation. There is little detailed analysis of this question. A general inhibition of either Tgfb or BMP would be expected to drive endothelial activation, not necessarily quiescence. A detailed analysis of the phosphorylation status of R-smads* in vivo *would seem important to understand this.*

*Overexpression of just the inhibitory smads may not at all reproduce the situation the authors find* in vivo.In conclusion, the present work provides an interesting starting point, with data that may become valuable for the field with time, once further functional validation has been done. It would be unreasonable to ask for full validation of the extensive regulatory networks. But it may be possible for the authors to further validate the role of Tgfb pathway, in particular to dissect whether the assumed quiescence regulation is indeed only achieved through epigenetic regulation of the inhibitory smads, or not (ligand levels etc.). If so, it would be critical to investigate whether such a mechanism provides endothelial cells with a state of resistance to ligand induced activation? Can the authors modify the methylation experimentally?

We sincerely appreciate the reviewer’s overall positive assessment of the manuscript. In the revised manuscript, we evaluated the potential systemic mechanism of quiescence by analysing TGFs family signaling in different organs in more detail. In addition to lung, we have included brain and heart in our analyses, showing the expression of Smad6 and Smad7, the phosphorylation status of receptor-regulated R-SMADs and TGFs family ligand levels (see Figure 5 and Figure 5—figure supplement 1). These data demonstrate that although ligand levels are high in adult tissue, R-SMAD phosphorylation is mostly decreased in EC, which might be due to Smad6 and/or Smad7 expression resulting in TGFs family signaling being repressed in young adult compared to infant tissues. Notably, cellular experiments of enforced expression of SMAD6 and/or SMAD7 in HUVEC demonstrated reduced R-SMAD phosphorylation upon ligand stimulation (Figure 5C), thereby confirming a certain state of resistance in EC expressing inhibitory SMADs.

The reviewer also asked if it would be possible to modify methylation experimentally. This is definitely an interesting suggestion. Yet, we don’t necessarily consider this of major relevance for the present scope of the study. Technically, CRISPR-Cas9-based epigenome technology has recently become available for the precise perturbation of the activity of specific regulatory elements (Klann et al., 2017; Hilton et al., 2015). We are planning on performing such experiments in the future but believe that this would be beyond the scope of the present manuscript.

As mentioned by the reviewer, mice were purchased from Taconic. Upon arrival, mice were kept in “Individually Ventilated Cages” (IVC) to maintain the health status as declared be Taconic as “Taconic Health Standard: Murine Pathogen Free™”. This information is now included in the manuscript (Subsection “Mice”).

The purity of isolated EC was validated after sorting as shown in Figure 1—figure supplement 1B, which is now explicitly stated in the main text (subsection “Endothelial transcriptomic changes during acquisition of vascular quiescence”). Furthermore, we included the results of additional EC population characterization experiments in Figure 1B and Figure 1—figure supplement 1C. As the expression of Notch3 was specifically mentioned by the reviewer, we performed qPCR to validate the expression to be clearly detectable in infant lung EC (Figure 2—figure supplement 1A). As the RNA-seq data pointed to different expression level of the different Notch receptors, we aimed at getting a more precise picture of the expression of the different Notch receptors in EC during acquisition of vascular quiescence. Therefore, we also validated Notch1, *Notch2* and Notch4 expression by qPCR in independent biological EC samples. This analysis confirmed the higher expression of Notch1 and Notch4 (Δ Ct [Notch1 – Actin] = 5.4 ± 0.4, Δ Ct [Notch4 – Actin] = 3.8 ± 0.6) compared to *Notch2* and Notch3 (Δ Ct [*Notch2* – Actin] = 8.2 ± 0.9, Δ Ct [Notch3 – Actin] = 8.1 ± 2.0). The results of these analyses are shown in Author response image 1.

**Author response image 1. respfig1:** Expression of Notch receptors in lung EC. qPCR analysis was performed in independent biological samples. n≥3; mean ± SD..

Reviewer #2:
*This study provides a description of the transcriptomic and epigenetic landscape of endothelial cell maturation by performing analyses on ECs isolated from day 8 and young adult mice (8-10 weeks old). Numerous changes are cataloged. Robust changes are seen in Smad6 and Smad7 and their roles as anti-angiogenic, anti-proliferative signaling molecules is validated by* in vitro studies.A detailed analysis of Smad6 and 7 on control of EC responses to BMP and lateral branching was published last year by Bautch's group. It is not cited here.In the Discussion section it is stated that "SMAD6 and SMAD7 have originally been identified as vascular SMADs and are, in fact, expressed almost exclusively in EC (Topper et a., 1997)" citing a 20 year old paper. I doubt that this is true. A PubMed search suggests that they are ubiquitously expressed.

We sincerely appreciate the reviewer’s assessment of the manuscript. We fully agree with the reviewer and apologize for not citing the mentioned paper. This has been remedied in the revised manuscript. The Mouillesseaux et al., 2016 reference has been included in the Introduction as well as in the Results section. The reviewer doubts our historical reference that SMAD6 and SMAD7 have been identified as vascular SMADs. Both molecules are, of course, ubiquitously expressed. Yet, the title of the original Topper et al., 1996 reference reads: “Vascular MADs: Two novel MAD-related genes selectively inducible by flow in human vascular endothelium”. And despite their ubiquitous expression, the Gimbrone group was probably not far off as portraying them as vascular SMADs considering that they exert particularly prominent vascular functions, e.g., as reflected by the phenotype of the knockouts. In revising the manuscript, we have spelled this our more clearly (Discussion section).

Reviewer #3:The manuscript seeks to investigate the transcriptional and epigenetic mechanisms that control endothelial quiescence. This is still a fairly unexplored area of vascular biology; thus, the study is timely and welcome. To achieve their aim, the Authors use FACS sorted lung EC from newborn mice (P 8) and young adult mice (8-12 weeks) for RNA Seq and methylation analysis.The study takes a novel approach to the question of quiescence. Transcription profiling identifies over 2000 genes differentially expressed in newborn vs young adult; GO analysis identifies genes involved in multiple families including cell cycle, and proliferation analysis of EC from various organs using EdU and Ki567 profiling shows decreased EC proliferation in the adult. Figure 1 shows two of the pathways that may be involved in this phenotypic switch, namely the JAK-STAT pathway and the TGFb pathway. Given that the focus of the study is on quiescence, it would be important to provide the list of genes involved in cell cycle regulation. The network analysis in Figure 1—figure supplement 2 is not very helpful; it could be removed and replaced by small selected networks with each gene labelled.Of the partial list of genes involved in angiogenesis shown in Figure 2, the very noticeable absents are members of the VEGF/VEGFR families. The authors acknowledge this, but do not discuss the possible reasons. This should be addressed in the Discussion section. Also, this somewhat weakens the argument made throughout the paper that angiogenic pathways are regulated during the progression to adulthood. A key player in this process appears to be TGFb, which becomes the focus of the second half of the manuscript. Therefore, I would recommend altering the sections of the paper where they claim that angiogenic pathways are crucially regulated. The title also should be modified, since it creates an expectation of a major focus on angiogenesis pathways, when in fact most of the validated analysis is on TGFb pathways.Having profiled the transcriptome, the authors proceed to profile the methylome. This is an interesting and novel experiment, and the result of potential value; other studies on EC methylation have been carried out but not on this model of young vs adult mouse EC. Data description and representation should provide more detail in the text and not only in the figure legends and Materials and methods section. Figure 3A shows a difference on CpG methylation and DMRs between newborn and young adult EC: is the difference statistically significant? Analysis of the location of DMRs identifies regions distal from the TSS as enriched in DMRs; the figure also shows EPU: enhancer promoterUnits, (defined in the figure legend not in the text): how are these identified?Analysis of TF motifs within DMRs identifies FOX and ETS as the most represented and conclude that FOX motifs are demethylated in adult whilst ETS motifs acquire methylation in adult EC, suggesting that ETS pathways, active in newborn mice, are switched off in the adult, and the opposite for FOX. Based on these findings, the Authors conclude that there is a switch between FOX and ETS pathways which controls the progression to adult quiescent EC. This is an interesting hypothesis, but it is not demonstrated and therefore the conclusions are too speculative. Firstly, the ERG (ETS) binding site is significant in both gain and loss regions – this is not mentioned in the text. Also, levels of TF from these families do not support the "switch" model, since they find no difference in the FOX group, an increase in 3/5 IRF TF and a mix of up-regulation, down-regulation or no effect in the 5 ETS factors investigated. Thus, motif analysis is not consistent with TF expression; that may not necessarily be a conflict, since there are more factors in each family, and their expression levels may not be the key determinant of function. However, lack of correlation between TF expression and TF motif methylation somewhat weakens the theory of a "switch" and suggests more of a modulation. There is also an issue with methylation status, which on a gene to gene basis does not consistently predict induction or repression of the gene (Figure 5D and Figure 6D). Also, are any FOXO or ETS target genes differentially regulated between infant and young adult mice (c.f. Figure 2A)? this is not mentioned, and it seems an odd omission to the study. Crucially, validation of the role of this FOX – ETS switch needs to be carried out experimentally, by testing whether the changes in DMRs at these motifs are transcriptionally meaningful and result in differential gene expression of the TF targets, as well as differences in EC proliferation.

*The study then moves on to the TGFβ pathway and SMADs signalling, with some straightforward* in vitro *validation of selected targets. Surprisingly, the TF motif analysis there is no sign of SMAD motifs. This is not discussed – it would be interesting to know the Authors view on this. Involvement of TGFb in maturation/differentiation is not a particularly novel finding, but it is still novel in this model and with this approach.*

There are two separate stories in this paper, which at present do not sit comfortably together: a more preliminary one, to do with the FOX/ETS switch and the other one, more mature, on the TGFb pathway. In summary, this is a very interesting study with novel data of potential interest to the vascular biology community. However, some of the conclusions are not supported by the data and require more validation.

We sincerely appreciate the reviewer’s overall positive assessment of the manuscript. We fully agree with the reviewer that the regulation of cell cycle genes is of particular interest in a study focusing on the transition of angiogenic EC to quiescent EC. We have therefore included a detailed gene list in the revised manuscript (Figure 1—source data 1). In addition, we validated the differential expression of three cell cycle regulatory genes in independent biological replicates of lung, brain and heart EC by qPCR (Figure 1D and Figure 1—figure supplement 1E).

We also agree with the reviewer that the statement of the regulation of angiogenic pathways is overstated. Following the reviewer’s advice, the revised manuscript focusses on TGFs family signaling and further validates TGFs family signaling molecules as key regulators of vascular quiescence. Correspondingly, the title has been modified as follows: “Genome-wide profiling of endothelial cells identifies TGFß family signaling as key regulator of vascular quiescence”. Furthermore, the interactome network analysis has completely been moved to the supplementary data (Figure 1—figure supplement 2) and we decided to leave out the FOX/ETS switch (please see also response to editor’s comment). As correctly noted by the reviewer, the absence of prominent regulation of VEGF/VEGFR signaling during acquisition of vascular quiescence needs further discussion. Therefore, the discussion now includes a new section dealing with this finding (Discussion section).

In addition to the complete re-phrasing of the discussion, we also extensively revised the Results section to include more detailed descriptions, specifically in the part describing the DNA methylation analysis. For instance, we included statistics demonstrating a significant increase of CpGs with >80% methylation level in adult EC compared to infant EC (p=6.87x10-6) (Figure 3A) (subsection “Vascular quiescence is accompanied by a prominent loss of DNA methylation at intronic enhancer regions”). EPU (enhancer-promoter units) were among the genomic regulatory features that helped to further characterize the quiescent-specific DMRs (Figure 3F). These units were defined by the Ren lab (Shen et al., 2012) “to gain a better understanding of enhancer/promoter organization”. They “assessed the correlation of the chromatin state at enhancers and polII occupancy at promoters for each possible pair of elements along a chromosome” and “observed that co-regulated promoters and enhancers tend to form clusters with variable sizes.” Thereupon, they “developed an algorithm to detect these local clusters, defined as enhancer–promoter units (EPUs).” We believe that the re-structuring and re-phrasing of the manuscript has substantially improved the readability and understanding of the presented findings.

[Editors' note: the author responses to the re-review follow.]

Summary:The aim of this revised manuscript from Schlereth et al., manuscript is to discover which molecules drive endothelial cell (EC) quiescence. To achieve this, they compare EC gene expression changes that occur as new born mice progress to adulthood and establish and maintain a quiescent adult endothelium. The group undertake an "-omics" comparison of FACS sorted (CD34+CD31+) lung ECs harvested from suckling mice (8-10dpp) versus those harvested from adult mice (6-8 weeks dpp) mice, including transcriptomic profiling and DNA methylomics.One question that came up in the discussion was in regard to the lack of Smad6 and 7 Western blots (or in situ or immunostaining in vivo) to look at the level of expression of these genes/proteins. Was it attempted and not successful? Two reviewers thought that it would be a very nice addition to the paper.

We sincerely appreciate this comment about presenting protein expression data of SMAD6 and SMAD7. As the reviewers, we felt the need to not only present RNA expression but also protein expression data. We tried hard to identify SMAD6 and SMAD7 in immunoblots or immunostainings. We tested towards this end several antibodies from different companies in immunoblots (Abcam, Life Technologies, R&D systems, Santa Cruz, Σ; published one from Cell Signaling Technologies #9519 was not available anymore). However, when we depleted SMAD6 and SMAD7 by shRNA, we could not detect a decline in protein bands that would reflect the qPCR data (specific and reliable Taqman assays) generated from exactly the same samples (Author response image 2) suggesting that the detected bands do not necessarily reflect endogenous SMAD6 or SMAD7, respectively. Yet, when we exogenously expressed SMAD6 in HUVEC the corresponding antibody detected a distinct band in the expected samples. At the height of this specific band no endogenous protein could be detected in our hands. Furthermore, even though qPCR data proved a high exogenous SMAD7 expression this could not be mirrored in immunoblots. In summary, the SMAD6 antibodies detected the exogenous SMAD6 but not the endogenously expressed protein and SMAD7 was not detected at all by any of the employed antibody in our hands. Similarly, immunostainings of SMAD6 or SMAD7 did not result in specific staining in our hands.

**Author response image 2. respfig2:** HUVEC were either depleted for (**A, B**) or overexpressing (**C, D**) SMAD6 and/or SMAD7. Both, shRNAs or overexpression constructs were introduced by lentiviral transduction followed by selection with puromycin (P) or neomycin (N). Panel A and C represents Taqman-based qPCR data and panel B and D the corresponding protein expression data from the same treatment replicates. SMAD6 antibody from Life Technologies (PA1-41026), SMAD7 antibody from R&D systems (MAB2029).

Reviewer #1:The manuscript by Schlereth et al., tackle the important yet unanswered question in the field of vascular biology, the acquisition of quiescence in endothelial cells (ECs). The authors compared freshly isolated ECs from infant and young adult mouse lungs to gain insight about the dominant regulators that generate stable vasculature in healthy, normal mice. By incorporating unbiased whole genome approach with transcriptome and methylome analysis, and also by analyzing ECs from multiple organs, this study provides valuable insights and information which has broad applicability for future studies in the field. The convergence of evidence from transcriptome analysis and methylome analysis all pointing toward TGFβ signaling and in vitro validation of the role played by Smad6 and Smad7 on EC quiescence make a strong case for TGFβ being a key regulator of EC quiescence. Few minor adjustments would help improve this manuscript before publication.1) Subsection “TGFs family signaling is inhibited in quiescent EC”: "Surprisingly, members of the VEGF/VEGFR pathway were not identified as being transcriptionally regulated." This sentence is rather confusing, given that in Figure 2A, heatmap showed upregulation of Vegfa in EC of young adult, and also in Figure 1—figure supplement 1B, EC of young adult seems to have higher expression of Kdr (Vegfr2) compared to EC of infant. Does it mean that heatmap and RT-PCR results are not statistically significant? Or does it mean that IPA analysis of upstream regulators did not reveal VEGF/VEGFR pathway as one of significantly altered signaling pathways? Also, in the Discussion section, the authors wrote "Since the expression of the primary VEGF signal transducing receptor Kdr/Vegfr2 was not changed,…". Is the expression level of Kdr not changed in RNA-seq or RT-PCR data, or both? More detailed explanation is required to avoid confusion.

We thank the reviewer for this constructive comment and apologize for the confusion. In fact, the upregulation of Vegfa (shown in heatmap of Figure 2A) is the only significantly differentially expressed VEGF pathway molecule that we identified. The Kdr expression difference between infant and young adult EC was neither in RNAseq data nor in qPCR data significant. Moreover, neither Reactome overlap analysis nor IPA upstream growth factor analysis identified the VEGF/VEGFR signaling pathway as being significantly differentially regulated. To avoid confusion, we deleted the mentioned sentence in the Results section, included a more detailed figure legend of Figure 2A, and rephrased the sentence in the Discussion section into “Since the expression of the primary VEGF signal transducing receptor Kdr/Vegfr2 was neither in RNA-seq nor in qPCR data significantly changed […]”.

2) Subsection “Clustered intragenic loss of methylation regulates TGFβ family and semaphorin signaling in quiescent endothelial cells”: "The correlation of quiescence-related expression and methylation profiles revealed an upregulated expression of genes with hypomethylated DMRs in adult EC independent of the genomic loci of one or several DMRs." This is clearly not the case for Sema6a as shown in Figure 4 and Figure 4—figure supplement 1A. Intragenic regions of Sema6a showed multiple DMRs, but its expression level is downregulated in young adult ECs. Smad7, on the other hand, showed multiple DMRs in intragenic regions and showed significant upregulation in expression level in young adult ECs, agreeing with the authors' claim. This discrepancy clearly suggests that hypomethylation does not guarantee upregulation of expression level of the affected genes. The authors should at least discuss this matter in more detail and avoid writing generalized statements concerning hypomethylation and the direction of gene expression changes, which seem oversimplified in the current version of the manuscript.

We thank the reviewer for this comment and apologize for not phrasing the correlation of transcriptomic and epigenetic data precisely enough. The reviewer is correct when stating that hypomethylation does not guarantee gene upregulation at the moment of analysis and refers to Sema6a as a bona fide example. This gene is significantly repressed in adult compared to infant mice while harboring multiple hypomethylated DMRs in adult EC. This suggests that the gene is repressed by other epigenetic mechanisms (histone modification, lincRNA interaction, etc.). Yet, when taking the complete genome-wide profiles into account, we see an excellent correlation of hypomethylation of the gene locus and gene induction in adult EC (see also overlap analysis shown in Figure 4F). For clarification, the corresponding sentences have been rephrased to better reflect the type of analysis (enrichment analysis) that was performed by using Gene Set Enrichment Analysis (GSEA) to correlate gene expression profiles without setting thresholds that might introduce bias (see subsection “Clustered intragenic loss of methylation regulates TGFβ family and semaphorin signaling in quiescent endothelial cells”).

3) Does the number of DMRs in a given gene have a direct correlation with the extent of expression change of that gene? In the Discussion section, the authors wrote "The presence of clustered DMRs was even more likely associated with differential expression changes of the affected gene, suggesting that we have identified intronic regions with high regulatory potential during the acquisition of vascular quiescence." However, heatmap in Figure 4—figure supplement 1B showed strong expression changes of Tek, which has only one DMR. Several other genes also showed significant expression changes while having a small number of DMRs (Ltbp3, Nid1, Tgfbr2, etc.). If there is no direct correlation, what is the meaning of Smad6 and Smad7 containing the largest cluster of DMRs? Are those just a lucky coincidence or are there more profound biological meaning to it? More detailed explanation would definitely help readers to understand this manuscript better.

We thank the reviewer for this constructive comment. The mentioned sentence from the discussion refers to the overlap analysis shown in Figure 4F. Relating to this figure panel, we state in the Results section that “we identified an increasing overlap of transcriptome and epigenome, when scoring genes on the basis of intragenic DMRs and even more so when scoring based on clustered DMRs (Figure 4F). This correlation implies that genes with intragenic clusters of loss of methylation DMRs are likely to be induced during acquisition of vascular quiescence, while those genes with clustered gain of methylation DMRs are likely to be repressed.” Certainly, we cannot exclude that for some genes this correlation does not hold true.

The inhibitory SMADs, Smad6 and Smad7, were most interesting for us as (1) they contained large clusters of hypomethylated DMRs in adult EC, and as (2) inhibition of TGFß family signaling was obvious from the transcriptome analysis. Thus, the methylome data gave us a hint how the inhibition of TGFß signaling might be achieved in adult endothelium.

Reviewer #2:The manuscript has significantly improved in terms of focus and clarity. There are a few aspects which in my view could be improved.1) Several pathways are mentioned, and some are validated in the course of the study. It is not always clear how these were prioritised. Early on, they state that "Functional enrichment analysis of the EC quiescence interactome led to the identification of JAK‐STAT signaling being highly enriched". Figure 1—source data 1 lists several other pathways, as expected. The text (subsection “Endothelial transcriptomic changes during acquisition of vascular quiescence”) should be reworded to make it clearer that several pathways were identified (some of which somewhat surprising to this reviewer), and that the JAK-STAT was the most significant. Strangely, this pathway is not mentioned again and is not validated. Is there a reason for this?

We thank the reviewer for this critical comment and gladly elaborate on this in more detail. We focused on the TGFß family signaling pathway as both genome-wide profiling approaches – independently – revealed a prominent regulation of relevant signaling molecules during the acquisition of endothelial cell quiescence. On the one hand, RNA-seq analysis demonstrated differential expression of multiple TGFß signaling molecules and upstream analysis of the complete set of significantly regulated genes revealed TGFß1 as the most significantly inhibited growth factor in adult EC (see subsection “TGFß family signaling is inhibited in quiescent EC”). On the other hand, the functional annotation of the genes next to differentially methylated regions (DMRs) identified by tagmentation-based WGBS similarly pointed towards a significant enrichment of TGFß family signaling molecules containing DMRs (see subsection “Clustered intragenic loss of methylation regulates TGFβ family and semaphorin signaling in quiescent endothelial cells”). Notably, we identified large clusters of hypomethylated DMRs in some genes and the genomic loci of *Smad6* and *Smad7* contained some of the largest clusters (see subsection “Clustered intragenic loss of methylation regulates TGFβ family and semaphorin signaling in quiescent endothelial cells”). As these two genes were also identified in the transcriptome analysis to be significantly repressed during transition to quiescence, we focused further validation studies prototypically on SMAD6 and SMAD7 in human endothelial cells.

We are aware of the fact that this study led to the identification of genome-wide profiles that can and will serve as starting point for many further studies. The reason why we mentioned the JAK-STAT signaling pathway without further validation studies was that it further specified the “immune system regulation” identified in the REACTOME overlap analysis of upregulated genes in adult EC (see Figure 1C). The specification of a pathway relevant in immune system regulation and at the same time EC quiescence was not obviously anticipated and therefore we included these interactome results. We considered these findings of interest for the prospective reader of the manuscript since only few previous publications have referred to a role of this pathway in EC quiescence biology. As advised by the reviewer, the section referring to the pathways identified to be enriched in the interactome analysis was revised to include more information (see subsection “Endothelial transcriptomic changes during acquisition of vascular quiescence”). Furthermore, we validated the upregulation of Stat1 during the acquisition of vascular quiescence in independent biological samples (Figure 2—Figure supplement 1A).

2) Figure 1—figure supplement 2 is not particularly useful, given that no nodes have been assigned labels (given the size of the network, that would be impossible). It is not easy to find a reader-friendly way to present this kind of analysis. It might be more helpful to select a few networks, including the Jak-Stat, Notch and TGF b pathways, and show network analysis related to these in detail, with names and direction of change in infant vs adult EC.

We thank the reviewer for this suggestion. As mentioned by the reviewer, the visualization of this kind of analysis is rather difficult. We included the interactome scheme to give the reader an impression of this kind of protein-protein interaction analysis. As suggested by the reviewer, we now also included cluster analysis of the quiescence-dependent interaction network which results in the identification of sub-communities of genes (see Figure 1—figure supplement 2C). These clusters were furthermore associated with biological function when enriched pathways could be identified. The corresponding results and methods section have been adjusted correspondingly (see subsection “Endothelial transcriptomic changes during acquisition of vascular quiescence” and subsection “Interactome analysis”).

3) The Authors move on to a selection of angiogenic molecules (Figure 2). How were these genes selected? Was the list manually curated or did they use DAVID or another GO software to select angiogenic regulators?

We thank the reviewer for this question. To establish this list, we made use of both curated gene sets of consortia listed in MSigDB and also manually selected additional molecules that are known to impact angiogenesis.

4) They eventually focus on TGFb. In subsection “TGFs family signaling is inhibited in quiescent EC”, they state: "upstream analysis of the complete set of differentially expressed genes including activation state prediction revealed TGFß1 as the most significantly inhibited growth factor". How does this relate to the pathways shown in Figure 1—source data1? the process which led them to focus on this pathway, in the context of other pathways identified here, should be more clearly explained

We thank the reviewer for careful inspection of the functional annotation analysis performed on the differentially expressed genes. The pathways shown in Figure 1—Source data 1 were derived from the interactome analysis which is based on protein interactions established from a seed set of proteins (differentially expressed genes during endothelial cell quiescence). In contrast, upstream analysis identifies regulatory molecules that impact the expression of a given gene (like for example a common upstream transcription factors that regulate a diversity of genes). Therefore, the pathways identified by means of these two different analyses do not necessarily overlap. Yet, TGFß family signaling was identified independently in both analyses. The reason why we further focused on TGFß family signaling has been outlined in more detail in comment 1 of reviewer #2.

5) How were DMR assigned to genes? Subsection “Clustered intragenic loss of methylation regulates TGFβ family and semaphorin signaling in quiescent endothelial cells”: "As enhancer regions could influence the expression of distant genes, up to three nearby genes were assigned to the dependent DMRs to identify genes likely affected by differential methylation". Please provide reference related to assignation method.

We apologize for not mentioning the tool used for the assignment of genes to the DMRs. This has been included in the revised manuscript (subsection “Clustered intragenic loss of methylation regulates TGFβ family and semaphorin signaling in quiescent endothelial cells”).

6) The correlation of quiescence-related expression and methylation profiles is an important aspect of this study, since it confirms the hypothesis that the cells' phenotype, and thence the infant to adult transition, is controlled by DNA methylation. Therefore, some of the data in Figure 4—figure supplement 1 should be in the main paper and should be described in more detail. There is a lot of information in this figure, which should be presented and not just briefly described in Figure legend. Also, Subsection “Clustered intragenic loss of methylation regulates TGFβ family and semaphorin signaling in quiescent endothelial cells”: "these data suggest that vascular quiescence is accompanied by a prominent loss of methylation in intragenic DMR clusters preferentially affecting and regulating the expression of genes involved in the regulation of TGFβ family, semaphorin and NOTCH signaling in this process". Is there a GO analysis showing how these pathways rank compared to others? It is interesting to focus on these pathways, but it should be clear whether these are the top pathways or whether they have been selected because of their functional relevance.

We thank the reviewer for this comprehensive comment. As suggested, we reviewed Figure 4 and Figure 4—figure supplement 1 thoroughly and added more detailed descriptions to the Results section. Figure 4 now contains two new panels that have previously been depicted in the supplement and that highlight the identification of DMR clustering in the DNA methylation analysis (new Figure 4B and C). Correspondingly, we added more information in the Results section to increase understanding and readability (see subsection “Clustered intragenic loss of methylation regulates TGFβ family and semaphorin signaling in quiescent endothelial cells”).

The sentence that the reviewer is citing summarizes the functional annotation analysis of the genes closest to DMRs. It refers to the tables shown in Figure 4A listing the top gene sets that overlap with the GREAT gene list, which assigned genes to the DMRs identified during the acquisition of vascular quiescence. The gene set list is sorted according to FDR in ascending order. This analysis implies a significant overlap of (1) the genes next to loss of methylation DMRs with TGFß and semaphorin signaling, and (2) the genes closest to gain of methylation DMRs with Notch and developmental signaling. This has now been stated more clearly in the Results section (see subsection “Clustered intragenic loss of methylation regulates TGFβ family and semaphorin signaling in quiescent endothelial cells”).

7) EC differentiation and maturation, the acquisition of quiescence and their angiogenic potential are all aspects of overlapping processes; The Authors repeatedly use "angiogenesis" as the opposite of quiescence. The opposite of quiescence is proliferation, and indeed in the early part of the study the Authors measure the proliferation rate of the cells under investigation. But I don't think that it is correct to equate "proliferative" with "angiogenic", given that several stages of angiogenesis do not require proliferation.

We fully agree with the reviewer that the opposite of endothelial cell quiescence is not just angiogenesis. Yet, we have in the manuscript consistently used the term “*quiescence*” to refer to the non-proliferative, resting state of adult endothelial cells in comparison to actively proliferating infant endothelial cells (as also obvious from the functional validation shown in Figure 1B and corresponding Figure 1—figure supplement 1D). In this respect, endothelial cell proliferation may be considered the most robust readout of the process of angiogenesis.

8) The data on SMAD 6&7 overexpression (Figure 5 and supplementary) is interpreted as confirming a key role for these two in the acquisition of endothelial quiescence. However, over-expression of SMAD6 alone does not affect proliferation, and the combined over-expression results in decreased cell viability. The interpretation of these results should be more cautious.

We apologize for the confusion. It was always our aim to interpret the results obtained from in vitro validation experiments adequately. Therefore, we stated that the decreased cell viability was only obvious upon prolonged culturing (see subsection “Epigenetically-regulated SMAD6 and SMAD7 control vascular quiescence”). Moreover, we are aware of the fact that SMAD6 and SMAD7 do not alone mediate endothelial cell quiescence. Yet, the data presented in Figure 5 suggest that they are regulating important processes (proliferation and migration) involved in this process. The wording of the title was consequently not only referring to SMAD6 and SMAD7 but rather to the TGFß family signaling pathway at large. To avoid confusions generated by the title, we used the opportunity to change the title to better reflect the studies content: “The transcriptomic and epigenetic map of vascular quiescence in the continuous lung endothelium”. This title aims at raising the readers interest and guiding her/him to the abstract that contains a detailed data summary (please see also response 3 to reviewer 3).

9) The conclusions are that the TGFb family is critical for the process. But what happened to the STAT pathway which was highlighted in Figure 1? This should be mentioned in the Discussion section.

We thank the reviewer for this comment and gladly elaborated in more detail on the potential role of JAK/STAT signaling during acquisition of endothelial cell quiescence in responses to comments 1 and 2. As already mentioned above, we included the JAK/STAT findings since we considered them surprising as they may contribute to immune competence acquisition during adolescence. In turn, the primary focus of the manuscript is on mechanisms of endothelial cell quiescence as it relates to the switch from the proliferative, angiogenic to the quiescent, resting phenotype. As we believe that it would distract from the main focus of the manuscript if we were to expand the discussion on the JAK/STAT pathway, we added more details in the results part. Obviously, a genome-wide analysis will inherently yield many more interesting data that can be discussed

Reviewer #3:The aim of this revised manuscript from Schlereth et al. manuscript is to discover which molecules drive endothelial cell (EC) quiescence. To achieve this, they compare EC gene expression changes that occur as new born mice progress to adulthood and establish and maintain a quiescent adult endothelium. The group undertake an "-omics" comparison of FACS sorted (CD34+CD31+) lung ECs harvested from suckling mice (8-10dpp) versus those harvested from adult mice (6-8 weeks dpp) mice, including transcriptomic profiling and DNA methylomics. The authors should be congratulated on bioinformatics analysis and interpretation of a large body of genome wide expression data that is certainly correct and gives insights into the adult quiescent endothelial cell state. However, since e Life tends to be read by biologists, the text could be expanded slightly to make it easier for biologists with fewer quantitative skills to assess and follow.The extensive genome-wide unbiased data reveals that components of active canonical TGF β and BMP signaling are down regulated in the adult quiescent state, exemplified by low levels of expression of receptors for TGF β and BMP and low levels of phosphorylated pSmad2/3 and pSmad1/5/8. This is interesting, and consistent with the widely accepted view of TGF β signaling serving in homeostatic regulation of epithelial and endothelial growth. The prediction would be that this signaling pathway would be upregulated when homeostasis is perturbed (e.g. by wounding, damage to endothelium, inflammation) in order to restore quiescence, a phenomenon that has been described extensively in epithelia and to a lesser extent for endothelia, but not addressed in the current study.The major novel finding of the current study is that genes encoding the inhibitory SMADs, Smad6 and SMAD7, are released from the epigenetic suppression that exists in lung ECs of young mice, such that in the adult these genes show extensive DNA hypomethylation at intronic CpG clusters indicative of higher gene expression, and consistent with the observation of higher Smad6/7 transcript levels by RNAseq and TAQMan. Thus, a major difference between lung EC from suckling versus adult mice is that in the former Smad6/7 genes are epigenetically repressed, whereas in the adult excessive Smad6/7 expression suppresses active TGF β/BMP signaling. The authors also show that in brain and heart EC, there too is epigenetic suppression of Smad6/7 in suckling mouse lung EC and release from such suppression in adult lung EC, suggesting that this is a universal rather than tissue specific effect. The manuscript (a) provides a useful resource for individuals interested in analysis of EC gene expression, (b) helps to place into context earlier findings of contextual actions of TGF β on EC, and (c) forms the foundation for further mechanistic studies.

We thank the reviewer for these positive comments, and the specific suggestions to further improve the readability of the manuscript. In fact, the reviewer’s comments mostly pointed in this direction and helped us to work on the general comprehensibility. Therefore, we believe that the revised manuscript is easier to read and to understand for the broad audience of the journal and not only to the specialist reader.

The previous reviewers asked for more in vivo validation of data. Although the authors go some way to provide this in Figure 5, the data in this figure is from total tissue lysate not from ECs, and would be include immune cells, epithelial cells and fibroblasts. It would be important to demonstrate that the high Smad6 and Smad7 transcript levels in adult versus young lung ECs are translated in to differential protein levels, either by Western and/or IHC.

We thank the reviewer for approaching the issue of SMAD6 and SMAD7 protein detection. We kindly refer to our response to the editor’s comment

If not already assessed in the literature or by the authors, future studies should address whether Smad6/7 protein is diminished in EC during pathological angiogenesis? NB Smad6 has at least as high expression in smooth muscle cells and pericytes, so it would be important to differentiate these two populations from EC.

We thank the reviewer for this comment on the role of SMAD6/7 in disease. Obviously, we were curious if published data would reveal important roles of the mentioned genes in pathological angiogenesis. Some studies have implicated SMAD6 and SMAD7 as important molecules in the regulation of TGFβ family activity in human diseases. For example, most cancer studies point towards a tumor suppressive function of SMAD6, as SMAD6 negative cancer show poor survival compared to SMAD6 positive cancer (Mangone et al., 2010; Osawa et al., 2004). In line with this, SMAD6 has been demonstrated to be downregulated in lung adenocarcinoma tissue compared to normal controls (Frullanti et al., 2012). Notably, the repression of both, SMAD6 and SMAD7 in tumors has been contributed to epigenetic silencing by promoter or gene body CpG hypermethylation (Bjaanaes et al., 2016; Hu et al., 2014; Mullapudi et al., 2015). Yet, these data are generated from whole tumor/organ tissue not differentiating between the different cell populations. There is also one study investigating monocrotaline-induced pulmonary arterial hypertension rat model (disordered proliferation of EC) that showed a reduction in Smad6 and Smad7 expression (Morty et al., 2007). Future work in our lab will consequently include pathology-related experiments in SMAD6/7 ECKO mice.